# DIFFERENCE-AWARE RETRIEVAL POLICIES FOR IMITATION LEARNING

**Quinn Pfeifer**[1], **Ethan Pronovost**[1], **Paarth Shah**[2], **Khimya Khetarpal**[3,4], **Siddhartha Srinivasa**[1], **Abhishek Gupta**[1,2]

[1]Paul G. Allen School of Computer Science & Engineering, University of Washington
[2]Toyota Research Institute
[3]Google DeepMind
[4]Mila

## ABSTRACT

Parametric imitation learning via behavior cloning can suffer from poor generalization to out-of-distribution states due to compounding errors during deployment. We show that reusing the training data during inference via a semi-parametric retrieval-based imitation learning approach can alleviate this challenge. We present **D**ifference-**A**ware **R**etrieval **P**olicies for Imitation Learning (**DARP**), a semi-parametric retrieval-based imitation learning approach that addresses this limitation by reparameterizing the imitation learning problem in terms of local neighborhood structure rather than direct state-to-action mappings. Instead of learning a global policy, DARP trains a model to predict actions based on $k$-nearest neighbors from expert demonstrations, their corresponding actions, and the relative distance vectors between neighbor states and query states. DARP requires no additional assumptions beyond those made for standard behavior cloning – it does not require additional data collection, online expert feedback, or task-specific knowledge. We demonstrate consistent performance improvements of 15-46% over standard behavior cloning across diverse domains, including continuous control and robotic manipulation, and across different representations, including high-dimensional visual features.

## 1 INTRODUCTION

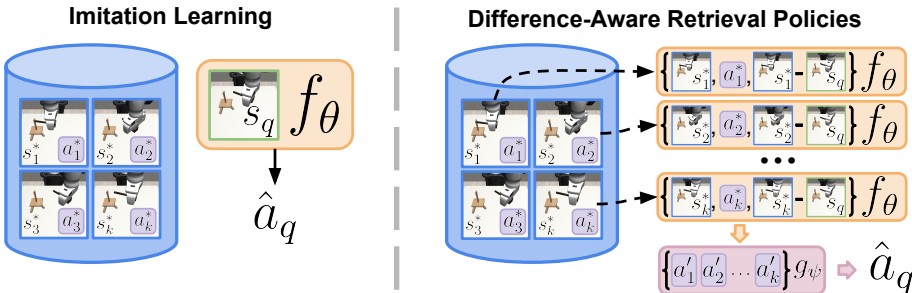

Figure 1: **Overview of DARP:** Unlike standard BC (left), DARP (right) utilizes a retrieval-based reparameterization centered around difference vectors between query states and retrieved neighbors. In standard behavior cloning, the dataset of expert state-action pairs is used only for training and is discarded at inference-time, while DARP utilizes it to performs retrieval to find a local neighborhood of expert state-action pairs around each query point $s_q$.

Imitation learning via behavior cloning (BC) (Pomerleau, 1991) has enabled robots to learn complex, dexterous behaviors from expert demonstrations (Zhao et al., 2023; Chi et al., 2024; Black et al., 2024; Chung et al., 2014). Yet despite its simplicity, BC often proves brittle in practice, especially for long-horizon tasks (Ross et al., 2011). The core issue is *covariate shift*: small errors accumulate

during rollouts, driving the agent into states not well represented in the demonstration data (Spencer et al., 2021; Ross et al., 2011). In such out-of-distribution regions, BC policies are highly unstable, producing unreliable and high-variance behavior that frequently leads to failure.

This problem is well recognized, and many approaches have been proposed to mitigate compounding error (Ross et al., 2011; Venkatraman et al., 2015; Ke et al., 2024b; Levine et al., 2020). However, these typically go beyond the standard BC assumptions, requiring simulators, interactive experts, large quantities of sub-optimal data, or strong task-specific structure. By contrast, our goal is to remain in the pure BC regime: learn only from expert state–action pairs, with no additional supervision or feedback. The central question is thus: *can we reduce the variance of BC policies using only the original demonstration dataset?*

From a statistical standpoint, BC minimizes only the supervised risk on expert states. This controls bias on the training distribution, but leaves variance unchecked: in low-density regions of the state space (which are often encountered during closed-loop rollouts), the learned policy can oscillate arbitrarily. A natural remedy is to enforce *smoothness*, so that nearby states yield similar predicted actions (Kobayashi, 2022; Asadi et al., 2018; Ke et al., 2024a; Chen et al., 2024). This discourages spurious fluctuations and improves rollout stability. Several approaches to encourage smoothness have been explored, see related work in Section 4.

Although sometimes effective, each has drawbacks: augmentation does not guarantee consistency, global priors can blur distinct behaviors, temporal penalties only act along time (not space), and explicit graph regularizers require tuning extra smoothness hyperparameters.

A complementary line of work contrasts global and local learning. "Global" supervised models (Black et al., 2024; Zhao et al., 2023; Chi et al., 2024) attempt to compress the entire demonstration dataset into a single parametric function, which is typically brittle under distribution shift. "Local" methods (Pari et al., 2022; Mansimov & Cho, 2018; Salzberg & Aha, 1994) instead adapt predictions to the structure of the dataset itself, consulting neighborhoods of similar states and generating outputs from non-parametric or semi-parametric operations on the training distribution of expert behavior. This locality offers robustness since it avoids reliance on a single parametric function, but it also has limitations: its effectiveness inherently depends on the distance metric, naive averaging of neighborhood can blur distinct actions and struggle to represent multimodality, and treating neighbors only in terms of their absolute states can limit generalization.

We introduce **D**ifference-**A**ware **R**etrieval **P**olicies for Imitation Learning (**DARP**), which combines the robustness of local methods with the stability of regularized global policy learning. At inference time, rather than predicting actions to execute only from the current query state via a feedforward pass on a parametric function, DARP (Fig. 1) first retrieves a set of $k$ neighbors from the training corpus and then predicts $k$ actions conditioned on tuples of (neighbor state, associated action, and difference from the query state). These neighbor-informed predictions are then aggregated in a permutation-invariant manner to produce a single robust action prediction. This design both grounds predictions in observed data (due to non-parametric retrieval) and implicitly enforces local consistency (due to parametric action prediction, conditional on the retrieved neighbors). We show that doing so reduces variance without requiring any additional assumptions beyond those made for standard Behavior Cloning – we need no additional data, online supervision, or task-specific knowledge. In spectral terms, this form of neighbor aggregation approximates a Laplacian filter on the $k$-NN ($k$-nearest neighbor) graph of expert states, providing a parameter-free form of smoothing that adapts to the local density and geometry of the dataset.

We provide both theoretical and empirical evidence that while operating under the same requirements as behavior cloning, DARP improves performance considerably by reducing variance and enhancing robustness to distribution shift. Our analysis formalizes the connection to Laplacian regularization, showing that DARP implicitly applies a fixed low-pass spectral filter that suppresses high-frequency variance. Empirically, on imitation learning evaluations, DARP achieves 15–46% gains over typical behavior cloning across continuous control (MuJoCo), robotic manipulation (Robosuite, Robocasa), and high-dimensional visual imitation tasks (Robosuite with image state). We demonstrate that DARP is a general, scalable architecture that naturally extends to image-based domains, with rich policy classes like transformers and Gaussian mixture models. We perform a careful set of ablations to highlight the importance of our particular choice of representation and architecture, providing general-purpose insights into retrieval-based algorithms for sequential decision-making problems.

## 2 DIFFERENCE-AWARE RETRIEVAL POLICIES FOR IMITATION LEARNING

In this work, we instantiate a new class of imitation learning methods that get the best of both "global" parametric learning methods and "local" learning methods. We propose a new architecture and simple training objective that allows for learning under the same requirements as typical behavior cloning, while providing significant improvements both theoretically and empirically. In this section, we thoroughly derive theoretical guarantees from first-principles. Readers primarily interested in the practical experimental results may skip to Section 2.5 for a self-contained summary of DARP before proceeding to Section 3 for empirical results.

As a warmup, we define the problem setting (Section 2.1) and discuss a variant of regularized imitation learning (Section 2.2) that imposes additional structure from the data for improvements in variance, generalization, and stability. In Section 2.3, we then show how the benefits of explicitly regularized learning can be implicitly accomplished by modifying policy *architecture* rather than the objective. Finally, in Section 2.4 we introduce our practical algorithm DARP, which realizes these benefits through a semi-parametric retrieval augmented architecture that can be generally applied to imitation learning with modern neural networks and generative modeling tools.

### 2.1 PRELIMINARIES: BEHAVIOR CLONING FOR IMITATION LEARNING

We operate in the typical imitation learning setting, formalized by a finite-horizon Markov Decision Process (MDP), $\mathcal{M} = \{\mathcal{S}, \mathcal{A}, P_0\}$, where $\mathcal{S}$ is the state space, $\mathcal{A}$ is the action space, and $P_0$ is the initial state distribution. A *policy* maps a state to a distribution of actions $\pi_\theta : s \to \Delta_\mathcal{A}$ so as to maximize task-relevant objectives. We assume access to expert human-provided demonstrations $\mathcal{D}^*$ as a collection of state-action pairs: $\mathcal{D}^* = \{(s_j^*, a_j^*)\}$. We use the notation $s^*$ and $a^*$ specifically to denote states and actions belonging to the expert dataset. The behavior cloning (Pomerleau, 1991) algorithm learns a policy $\pi_\theta$ from this dataset by casting imitation as a typical supervised learning problem — $\arg\max_\theta \mathbb{E}_{(s^*, a^*) \sim \mathcal{D}^*} [\log(\pi_\theta(a^* \mid s^*))]$. While the distribution class of $\pi_\theta$ can be an arbitrary complex generative model (Lipman et al., 2023; Chi et al., 2024), we will start with a Gaussian parameterization for the sake of simplicity[1].

### 2.2 WARM-UP: NEIGHBOR MANIFOLD REGULARIZED IMITATION LEARNING

While behavior cloning minimizes only the supervised imitation loss over expert states drawn from $\mathcal{D}^*$, such an objective alone does not control how the policy behaves on states that deviate from the manifold of expert states. In practice, accumulating errors lead the agent to out-of-distribution regions where a BC policy may act arbitrarily, especially for overparameterized neural networks (Ross & Bagnell, 2010).

To mitigate this, we note that behavior cloning enforces function evaluations only at the training states, but it does not explicitly take into account the relationship between states (and their corresponding actions) in a neighborhood, thereby ignoring the underlying data manifold. To incorporate this information into policy learning, let us consider a modified objective that introduces a regularization term that explicitly encourages *local consistency* of predictions: nearby states in the expert dataset should be mapped to similar actions. This intuition leads to the following neighborhood-regularized loss ($\mathcal{L}_{\text{MRIL}}$), where the standard imitation learning objective ($\mathcal{L}_{\text{BC}}$) is combined with an additional smoothness penalty ($\mathcal{L}_S$) enforcing predictions to respect the geometry of the dataset rather than relying solely on pointwise supervision.

$$\mathcal{L}_{\text{MRIL}}(f) = \underbrace{\mathbb{E}_{(s,a) \sim \mathcal{D}^*} \left[ \ell(f(s), a) \right]}_{\text{supervised risk}(\mathcal{L}_{\text{BC}})} + \lambda \underbrace{\mathbb{E}_{s \sim \mathcal{D}^*} \left[ \sum_{i \in \mathcal{N}_k(s)} w_i(s) \left\| f(s) - f(s_i^*) \right\|_2^2 \right]}_{\text{smoothness regularizer}(\mathcal{L}_S)}, \tag{1}$$

where $\ell(f(s), a)$ is the supervised imitation loss, $\mathcal{N}_k(s)$ are the $k$-nearest neighbors of $s$ from the expert dataset, and the weights $w_i(s)$ are normalized kernel weights based on the state differences - $w_i(s) \propto K_\Delta \left( \frac{\|s_i^* - s\|}{h} \right)$. As we discuss briefly below (and in detail in Appendix A.1.1), this corresponds

---

[1]We show that this can be relaxed in Section 2.4

to a form of *manifold regularization* or Laplacian smoothing, where the policy is penalized for high-frequency variation across the neighborhood of expert states. This manifold regularization provably leads to improvements in policy variance, stability, and generalization.

**Theorem 1** (Manifold Regularized BC ($\mathcal{L}_{\text{MRIL}}$) improves over vanilla BC ($\mathcal{L}_{\text{BC}}$)). *Let $f^* : \mathscr{S} \to \mathscr{A}$ be the true, underlying expert policy, assumed to be $C^2$-smooth on a compact state space $\mathscr{S}$. Let $f : \mathscr{S} \to \mathscr{A}$ denote the learned policy estimator. Consider two estimators trained on expert demonstrations:*

1. **Vanilla BC:** *a global supervised model minimizing*

$$\mathcal{L}_{\text{BC}}(f) = \mathbb{E}_{(s,a)\sim\mathscr{D}^*}[\ell(f(s), a)].$$

2. **MRIL:** *a neighbor-based estimator minimizing*

$$\mathcal{L}_{\text{MRIL}}(f) = \mathcal{L}_{\text{BC}}(f) + \lambda \mathbb{E}_{s\sim\mathscr{D}^*}\left[\sum_{i\in\mathscr{N}_k(s)} w_i(s)\left\|f(s) - f(s_i^*)\right\|_2^2\right],$$

   *where $w_i(s)$ are the kernel weights defined above and $\lambda > 0$.*

*Then, under the smoothness assumption on $f$, the following hold:*

(i) Variance reduction: *The Laplacian penalty in MRIL acts as a data-dependent Tikhonov regularizer, yielding smaller estimator variance than vanilla BC.*

(ii) Smoothness guarantee: *Minimizers of $\mathcal{L}_{\text{MRIL}}$ satisfy a uniform bound on the local Lipschitz constant of $f$, whereas vanilla BC admits interpolants with arbitrarily large Lipschitz constants between training states.*

(iii) Policy stability: *In a closed-loop rollout, the deviation recursion*

$$\Delta_{t+1} \leq L_s\Delta_t + L_a\|\pi(s_t) - f(s_t^*)\|^2$$

   *accumulates error linearly for vanilla BC, but sublinearly for MRIL, since the smoothness regularizer enforces $\|f(s) - f(s')\| = O(\|s - s'\|)$ for neighbors $s, s'$.*

*This suggests that MRIL enjoys strictly better generalization and stability guarantees than BC.*

*Proof sketch.* We defer the detailed proof to Appendix A.1.1, but provide a brief sketch. The key idea of the proof is to first show that the smoothness regularizer directly corresponds to a graph Laplacian penalty on a graph constructed by a $k$-nearest neighbor ($k$-NN) affinity matrix defined by the kernel $w_i$. Next, we show that as the number of samples tends to infinity, this graph Laplacian penalty converges to the weighted Dirichlet energy (Belkin & Niyogi, 2008; Zhou et al., 2003). Minimizing this Dirichlet energy (1) ensures that the learned $f$ is locally Lipschitz almost everywhere, ensuring smoothness and, in turn, policy stability, and (2) corresponds to Tikhonov regularization, thereby reducing estimator variance, while keeping the bias controlled. □

Intuitively, the smoothness regularizer is not merely penalizing pairwise disagreements between neighbors, but is driving the learned policy to be smooth with respect to the underlying data manifold. In particular, it shrinks the local Lipschitz constant of $f$ along directions where the data density $p(s)$ is high, ensuring that small changes in state lead to small, consistent changes in the predicted action. As a result, the policy generalizes more reliably on in-distribution (ID) states and extrapolates in a structured manner on new out-of-distribution (OOD) states in the neighborhood.

## 2.3 Implicit Manifold Regularization via In-Context Architectures

While our MRIL objective does amortize local learning to provide improvements over vanilla BC, there are two notable drawbacks. First, the presence of a hyperparameter $\lambda$ that must be tuned to balance supervised accuracy and smoothness. Second, the requirement to optimize a modified, regularized objective rather than a standard BC objective may modify the optimization landscape in

---

[2]$L_s$ and $L_a$ are the Lipschitz constants of the environment transition dynamics with respect to state and action, respectively. We denote $\pi(s_t)$ as the action predicted by the agent from the state at time $t$, $s_t$.

adverse ways. This raises a natural question: *can we obtain the same benefits conferred by MRIL (Eq. 1), by modifying the policy architecture rather than modifying the objective?*

In this section, we introduce a retrieval-based change in policy architecture that leads to an *implicit* manifold regularization effect (iMRIL), despite using a standard imitation objective. With iMRIL we can obtain the benefits of Laplacian smoothing (from MRIL) by training on a standard BC objective (as shown in Fig. 2), without introducing $\lambda$ as an additional hyperparameter for training. We then build on this algorithm to develop a practical instantiation of this method (DARP) in Section 2.4.

**iMRIL architecture:** The high-level idea behind iMRIL is simple – we propose moving the neighborhood aggregation (averaging) operation from the objective (as in Eq. 1) to the architecture itself. So instead of learning a standard feedforward predictor $f(s)$ that is trained against a neighborhood regularized smoothness objective (Eq. 1), we propose embedding the structure of neighborhood aggregation directly into the parameterization of the action predictor $\hat{f}$ itself, while maintaining the objective as standard imitation learning. iMRIL learns the

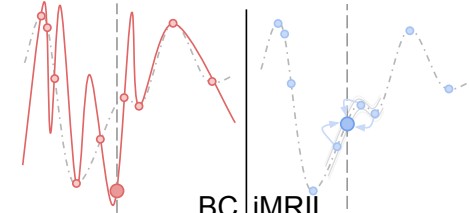

Figure 2: **iMRIL implicitly achieves Laplacian smoothing, which reduces variance and enforces local consistency**, whereas the lack of smoothness constraint on standard BC allows for arbitrarily jagged function approximations.

parameters of a per-state predictor $f_\theta$ such that an action predictor explicitly parameterized via neighborhood-aggregation $\hat{f}(s^*) = \frac{1}{k} \sum_{i \in \mathcal{N}_k(s^*)} f_\theta(s_i^*)$ across nearest neighbor states from the training set $\{s_i^*\}_{i \in \mathcal{N}_k(s^*)}$ generates accurate predictions of the corresponding expert action $a^*$. With this parameterization, iMRIL optimizes at training time:

$$\arg\min_\theta \mathbb{E}_{(s^*,a^*) \sim \mathcal{D}^*} \left[ \left\| \underbrace{\left( \frac{1}{k} \sum_{i \in \mathcal{N}_k(s^*)} f_\theta(s_i^*) \right)}_{\hat{f}(s^*)} - a^* \right\|_2 \right] \tag{2}$$

At deployment time, inference can be performed on a new state $s_q$ simply by retrieving the $k$-NN of $s_q$ from the training set and performing neighborhood aggregation $\hat{a} = \hat{f}(s_q) = \frac{1}{k} \sum_{i \in \mathcal{N}_k(s_q)} f_\theta(s_i^*)$. As we show in Section 2.4, the particular parameterization of $f$ is of crucial importance and plays a significant role in the empirical performance of iMRIL- leading to the development of DARP.

Intuitively, we are parameterizing the action predictor $\hat{f}$ as an aggregation of predictions at neighbor states from the training data $f(s_i^*)$, and then learning $f$. Supervising the post-aggregation function implicitly prevents any $f$ predictions from being arbitrarily non-smooth, conferring the benefits noted in Section 2.2. We prove a direct equivalence of iMRIL to the Laplacian regularization in Section 2.2.

**Equivalence between iMRIL and MRIL:** While we defer a full proof of formal equivalence between MRIL and iMRIL to the Appendix Section A.1, we state our main result and a proof sketch to this effect here.

**Theorem 2** (iMRIL is parameter-free Laplacian regularization for BC (MRIL))**.** *Consider the symmetric normalized $k$-NN graph Laplacian $L$ (defined in Section 2.2), with eigenpairs $\{(\mu_j, u_j)\}_{j=1}^n$, where $0 = \mu_1 \leq \mu_2 \leq \cdots \leq \mu_n \leq 2$.*

*The minimizers of the explicit MRIL objective (Section 2.2) and the implicit iMRIL objective (Section 2.3) have the following closed form expansions*

$$f_{\text{MRIL}} = \sum_{j=1}^n \frac{1}{1 + \lambda \mu_j} \langle a^*, u_j \rangle u_j \qquad\qquad \hat{f}_{\text{iMRIL}} = \sum_{j=1}^n (1 - \mu_j) \langle f, u_j \rangle u_j$$

*iMRIL 's neighbor aggregation step applies the fixed spectral filter $\phi_{\text{iMRIL}}(\mu) = 1 - \mu$ to the graph Laplacian $L$, preserving low-frequency modes and suppressing high-frequency modes. The congruence between $\hat{f}_{\text{iMRIL}}$ and $f_{\text{MRIL}}$ shows that iMRIL is equivalent to a built-in form of Laplacian*

*smoothing (MRIL) with effective $\lambda \approx 1$ in normalized units. Unlike explicit regularization, this implicit filter requires no additional hyperparameter tuning.*

*Proof sketch.* We defer full details to Appendix Section A.1.2. The explicit regularizer admits a spectral solution by diagonalizing the $k$-NN Laplacian, yielding a filter of the form $(1 + \lambda \mu)^{-1}$ on each eigenmode. The implicit objective can be expressed as neighbor aggregation $\hat{f} = Sf$ with $S = D^{-1}A$, the random-walk matrix, which has the same eigenvectors and applies the fixed filter $1 - \mu$. Intuitively, both act as low-pass filters on the graph: modes with small eigenvalues (smooth variation across the data manifold) are largely preserved, while modes with large eigenvalues (rapid, high-variance fluctuations between neighbors) are strongly damped. Thus iMRIL implicitly performs Laplacian smoothing, reducing variance and enforcing local consistency without needing to tune $\lambda$. □

Note that the implicit Laplacian smoothing view does not replace the need to learn a policy; rather, it constrains the class of functions that can be represented after aggregation. The neighbor-conditioned network $f_\theta$ learns how expert actions vary under local perturbations, proposing locally adapted actions for each neighbor. The aggregation operator then enforces variance reduction by smoothing these proposals across the neighborhood. In this way, learning provides accuracy by correcting local bias, while aggregation provides stability by controlling variance.

## 2.4 DIFFERENCE-AWARE RETRIEVAL POLICIES: A PRACTICAL INSTANTIATION OF iMRIL FOR IMITATION LEARNING

Given the conceptual framework of iMRIL, we instantiate a practical algorithm for large-scale imitation learning. We build on the objective outlined in Eq. 2 and instantiate a careful choice of (1) parameterization, (2) neighbor aggregation that leads to strong empirical performance.

### 2.4.1 DIFFERENCE-BASED PARAMETERIZATION OF $f_\theta$

The objective described in Eq. 2 leaves the parameterization and input representations of $f_\theta$ open to broad interpretation. We make the observation that the neighborhood aggregation should learn how expert actions vary under local perturbations. This suggests that $f_\theta$ should use knowledge of *differences* between a query state and a neighbor state to adaptively propose locally adapted actions for each neighbor. In **D**ifference-**A**ware **R**etrieval **P**olicies (**DARP**), instead of simply parameterizing $f_\theta$ by $f_\theta(s_i^*)$, we provide additional context about the optimal neighbor action $a_i^*$, as well as the *difference* between the query state and the neighbor state $\Delta s_i = s_i^* - s_q$; a predictor $f_\theta$ predicts an action candidate $a_i'$ for a query state $s_q$ and a neighbor $(s_i^*, a_i^*)$ using the difference information as - $a_i' = f_\theta(s_i^*, a_i^*, \Delta s_i = s_i^* - s_q)$.

Let $\mathcal{N}_k(s_q)$ be the index set of the $k$-nearest neighbors retrieved according to some distance function $d(s_q, s_i^*)$.[3] For generating predictions with DARP, we can then perform neighborhood aggregation (as outlined in Section 2.3) to predict an action for any query state $s_q$

$$\hat{a}_q = f_{\text{DARP}}(s_q) = \frac{1}{k} \sum_{i \in \mathcal{N}_k(s_q)} a_i' \tag{3}$$

$$= \frac{1}{k} \sum_{i \in \mathcal{N}_k(s_q)} f_\theta(s_i^*, a_i^*, \Delta s_i = s_i^* - s_q). \tag{4}$$

At training time, this can be used to define a straightforward imitation learning objective from the expert dataset $\mathscr{D}^*$:

$$\arg\min_\theta \mathbb{E}_{(s_q, a_q) \sim \mathscr{D}^*} \left[ \left\| \hat{a}_q - a_q \right\|^2 \right] \tag{5}$$

where we optimize for the parameters of the predictor $f_\theta$, minimizing the discrepancy between the predicted action $\hat{a}_q$ and optimal action $a_q$. Given the simplicity of the objective, any parameterization can be used for $f_\theta$, in our case, standard feedforward or convolutional neural networks. As we show

---

[3]In our work we use the Euclidean distance in a pre-trained embedding space, although other neighborhood functions are also applicable. We refer the reader to Appendix Section A.2.1 for a thorough discussion of design decisions in constructing neighborhood sets via retrieval.

in Section 3, this difference-based parameterization is crucial for performance. At inference time, we generate actions to execute by retrieving $k$-NN and performing inference through the neighborhood aggregation operation defined in Eq. 3.

### 2.4.2 GOING BEYOND LINEAR AGGREGATION

While the process of neighborhood aggregation thus far has been restricted to averaging over neighborhood predictions $\hat{a}_q = \frac{1}{k} \sum_{i \in \mathcal{N}_k(s_q)} a_i'$, this is a special case of a broader class of permutation-invariant aggregation functions $g_\psi(\{a_i'\}_{i \in \mathcal{N}_k(s_q)})$. For instance, $g_\psi$ could be parameterized with more expressive set-compliant neural models like the set transformer (Lee et al., 2019) or DeepSets (Zaheer et al., 2017). This suggests a generalization of the prediction model in Eq. 3 as $\hat{a}_q = g_\psi(\{f_\theta(s_i^*, a_i^*, \Delta s_i = s_i^* - s_q)\}_{i \in \mathcal{N}_k(s_q)})$. Besides benefits in expressivity, generalizing from a simple averaging operation to a parametric aggregation model $g_\psi$ allows for the representation of richer action distributions (e.g Gaussian mixture models (Pignat & Calinon, 2019) or diffusion models (Chi et al., 2024)) than the Gaussian distribution that is implicit to the $l_2$-regression objective defined in Eq. 5. Rather than predicting $\hat{a}_q$ directly, DARP can predict the parameters $\alpha$ of an action distribution $p(a_q; \alpha)$ – for instance the means, covariances, and weights for a Gaussian mixture model, or the score function for a diffusion model. This allows DARP to perform maximum likelihood training of multimodal action distributions rather than just unimodal $l_2$-regression:

$$\arg\max_\theta \mathbb{E}_{(s_q,a_q)\sim\mathscr{D}^*} \left[\log p(a_q; \alpha_\theta(s_q))\right], \text{ where } \alpha_\theta(s_q) = g_\psi\left(\{f_\theta(s_i^*, a_i^*, \Delta s_i = s_i^* - s_q)\}_{i \in \mathcal{N}_k(s_q)}\right) \tag{6}$$

Inference for a query state $s_q$ can be performed by sampling from $p(a_q; \alpha_\theta(s_q))$, constructing $\alpha_\theta(s_q) = g_\psi\left(\{f_\theta(s_i^*, a_i^*, \Delta s_i = s_i^* - s_q)\}_{i \in \mathcal{N}_k(s_q)}\right)$ from a set of neighbors retrieved at test time. We refer readers to Appendix Section A.3 for detailed training pseudocode.

### 2.5 SUMMARY

We show that encouraging local consistency when fitting a behavior cloning policy corresponds to manifold regularization (Laplacian smoothing) and provably improves variance, smoothness, and stability over standard BC (Theorem 1). Rather than adding an explicit regularization term (and its attendant hyperparameter $\lambda$) into our loss function, we show that we can induce smoothing implicitly by building neighborhood aggregation into the policy architecture itself (Theorem 2). Thus, we propose the following algorithm: at training time, for each expert state-action pair $(s_q, a_q)$, DARP retrieves the $k$-nearest neighbor states from $\mathscr{D}^*$, computes difference vectors $\Delta s_i = s_i^* - s_q$ from each neighbor to the query state, passes each neighbor tuple $(s_i^*, a_i^*, \Delta s_i)$ through a network $f_\theta$ to produce candidate actions, and finally aggregates these candidates via a permutation-invariant function $g_\psi$ to predict the final action. This is trained only with the standard imitation learning objective.

## 3 EXPERIMENTAL EVALUATION

Next, we evaluate DARP in order to answer three key questions: **Q1:** Can DARP consistently outperform standard behavior cloning?, **Q2:** Can DARP handle more complex state representation and action distributions?, **Q3:** How do different architectural components contribute to DARP's performance gains? We conduct experiments across multiple domains using low-dimensional state representations, high-dimensional image features, and diverse action representations. Our evaluation includes continuous control tasks (MuJoCo), robotic manipulation (Robosuite), and specially designed discontinuous environments that stress-test the neighbor-based approach.

### 3.1 BASELINE COMPARISONS AND TASK DESCRIPTIONS

**MuJoCo Tasks:** The MuJoCo (Todorov et al., 2012; Fu et al., 2020) tasks entail controlling various legged figures in multiple embodiments to achieve forward locomotion on a flat plane. These tasks include: Hopper (single-legged hopping robot), Walker (bipedal humanoid), Ant (quadruped), and HalfCheetah (biped).

**Robosuite Tasks:** The Robosuite (Zhu et al., 2020) tasks all entail a single robotic arm manipulating objects. In the Stack task, the goal is to put a smaller cube on top of a larger one. In the Threading task, the goal is to manipulate a thin, needle-like tool and insert it into a small ring. In the Square Peg task, the goal is to manipulate a square wooden block with a hole in the center and place it onto a square peg.

**RoboCasa Tasks:** The RoboCasa (Nasiriany et al., 2024) tasks all entail a single robotic arm manipulating objects in a randomized kitchen setting. In the Drawer task, the goal is to close an open drawer. In the Door task, the goal is to close an open microwave oven door. In the Stove task, the goal is to twist a knob to turn off a stove burner.

**Baseline Comparisons:** We compare DARP against a variety of baselines and ablations: (1) **R&P (Sridhar et al., 2025):** refers to directly taking the action corresponding to the nearest neighbor, (2) **LWR (Pari et al., 2022):** refers to performing locally weighted regression on retrieved neighbors, (3) **BC:** refers to standard parametric behavior cloning, (4) **REGENT (Sridhar et al., 2025):** refers to a transformer-based in-context learning method conditioned on retrieved neighbors, (5) **MRIL:** refers to the explicitly smoothed version of DARP outlined in Section 2.2.

## 3.2 CAN DARP CONSISTENTLY OUTPERFORM STANDARD BEHAVIOR CLONING? (Q1)

In this experiment, we evaluate DARP's core hypothesis on tasks with low-dimensional state representations, where the distance metrics between states are well-defined and interpretable. This evaluation spans locomotion tasks from MuJoCo and robotic manipulation tasks from Robosuite and RoboCasa with data generated with MimicGen (Mandlekar et al., 2023). In these experiments, the aggregation function $g$ is implemented as a simple average of all neighbor action predictions $a'$.

| Method | Hopper | Ant | Walker | HalfCheetah |
|---|---|---|---|---|
| R&P (Sridhar et al., 2025) | 711.82 ± 85.63 | -305.97 ± 76.42 | 419.18 ± 50.21 | -178.64 ± 29.75 |
| LWR (Pari et al., 2022) | 1703.78 ± 245.95 | 846.59 ± 216.06 | 1484.91 ± 356.54 | 1945.82 ± 567.26 |
| BC | 2313.65 ± 203.75 | 2376.20 ± 339.43 | 2658.40 ± 274.08 | 1063.23 ± 371.08 |
| REGENT (Sridhar et al., 2025) | 1819.39 ± 186.24 | -302.10 ± 146.67 | 507.01 ± 76.10 | 169.85 ± 63.10 |
| MRIL | 2793.63 ± 156.41 | 3869.08 ± 241.00 | 4370.96 ± 168.13 | 701.58 ± 195.08 |
| **DARP** | **3545.57 ± 3.54** | **4383.28 ± 266.37** | **4894.01 ± 75.12** | **5515.41 ± 841.33** |
| **DARP Set Transformer** | 2965.86 ± 103.08 | **4063.79 ± 218.80** | **4752.42 ± 109.23** | 3417.85 ± 764.57 |

Table 1: **Both DARP and DARP Set Transformer outperform other approaches across all domains.** Performance Comparison of DARP vs. BC and other baselines across MuJoCo Environments Using Low-Dimensional State. Scores reported are averaged across 100 independent trials with 95% confidence intervals.

We find that DARP demonstrates substantial improvements over standard behavior cloning across all tested environments. We observe performance gains ranging from 15-25% points in robotic manipulation tasks and significant score improvements in locomotion tasks (see Table 1, Table 3, and Table 2). We observe that purely non-parametric methods (R&P and LWR) perform poorly on these tasks, and while MRIL is nearly always able to get a score higher than vanilla BC, the highest scores on this suite of tasks are always achieved by our DARP architecture.

Given the changes introduced for the practical instantiation in Section 2.4, we evaluate whether DARP scales up to higher-dimensional input representations such as images.

| Method | Drawer | Door | Stove |
|---|---|---|---|
| BC | 54 | 29 | 28 |
| **DARP** | **85** | **45** | **43** |

Table 2: Comparing across RoboCasa Environments using low-dimensional state features. Scores are listed as success percentage. DARP outperforms BC.

| Method | Stack | Thrd. | Peg |
|---|---|---|---|
| R&P | 38 | 11 | 31 |
| LWR | 21 | 39 | 30 |
| BC | 47 | 37 | 46 |
| **DARP** | **72** | **63** | **62** |

Table 3: Comparing across Robosuite Environments using low-dim state features.

### 3.3 CAN DARP HANDLE MORE COMPLEX STATE REPRESENTATION AND ACTION DISTRIBUTIONS? (Q2)

**High-Dimensional Visual Input Representations**. To test the applicability of DARP beyond the regime of compact, low-dimensional states, we evaluate DARP on simulated robotic manipulation tasks using R3M image embeddings (Nair et al., 2022). This tests whether the neighbor-based approach remains effective when states are represented as high-dimensional feature vectors extracted from visual observations (see Table 4). Observe that, not only does DARP outperform standard BC, the

| Method | Stack | Thrd. | Peg |
|--------|-------|-------|-----|
| BC | 44 | 38 | 17 |
| **DARP** | **75** | **76** | **52** |

Table 4: Success rates (%) on vision-based Robosuite tasks.

average improvement, $\sim 35\%$, is actually higher than the average improvement on Robosuite tasks in low-dimensional state ($\sim 22\%$). Empirically, this means that DARP was better at adapting to complex, high-dimensional state representations than standard BC.

**Multi-modal Action Distributions**. We show that DARP can solve complex multimodal imitation learning tasks such as the Push-T environment over 20% better than behavior cloning. We defer details to Appendix A.2.2.

### 3.4 HOW DO DIFFERENT ARCHITECTURAL COMPONENTS CONTRIBUTE TO DARP'S PERFORMANCE GAINS? (Q3)

**Ablation Study:** To understand which components of the DARP architecture contribute most to its performance gains, we conduct a comprehensive ablation study examining each design choice, namely (1) standard DARP; (2) DARP, but without including the neighbor actions; (3) an ensemble of 10 BC agents; (4) DARP, but we choose random neighbors as opposed to using a distance metric; (5) DARP, but we take the L2 norm of the distance vector; (6) BC baseline, which is just the query state $s_q$; (7) DARP, but include just the query state rather than the distance vector between the query state and neighbor states; (8) DARP, but using a permutation-dependent (so **not** permutation-invariant) aggregator to combine all $a'$s. We report in Figure 3 the results of this systematic ablation.

The ablation study reveals that distance vectors and permutation invariance are crucial for DARP's success, while neighbor actions have a more modest impact. Random neighbor selection performs poorly, confirming that meaningful distance metrics informing neighbor selection are crucial. The permutation-invariant aggregation function $g$ proves critical, as permutation-dependent alternatives significantly degrade performance.

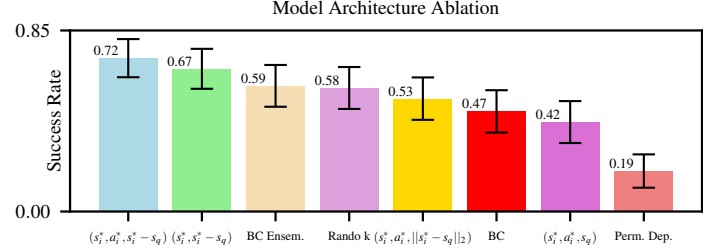

Figure 3: **Distance vectors and permutation invariance contribute heavily to DARP's success.** Exploration of how the performance of a DARP agent is impacted as various changes are made to the core architecture demonstrates that DARP success is most attributed to the distance vectors $(s_i^*, a_i^*, s_i^* - s_q)$. Success rate is averaged across 100 trials on the Robosuite Stack environment with 95% confidence intervals.

**Divergence Analysis:** To better understand DARP's success over standard BC, we analyze the point of divergence in rollouts in which the latter fails but the former succeeds. We identify the "step of divergence" as the point at which DARP and BC begin to receive a significantly different reward. We define $\tau_s$ and $\tau_\Delta$ as the 1st percentile of likelihoods of the training set (that is, 1% of the deltas seen at training time are less likely than $\tau_\Delta$). In all six different rollouts across two different tasks (the Robosuite Stack task and the MuJoCo Hopper task), the query state at the SoD has a state likelihood of $< \tau_s$ but a delta likelihood of $> \tau_\Delta$. This result bolsters our hypothesis that DARP gains occur partly due to improved prediction on slightly out-of-distribution states due to reparameterization in terms of difference vectors to neighbors. (see Figure 4 for plots of reward drift, SoDs, and state and delta likelihood for one task.) See Appendix A.2.4 for additional experiments regarding DARP robustness.

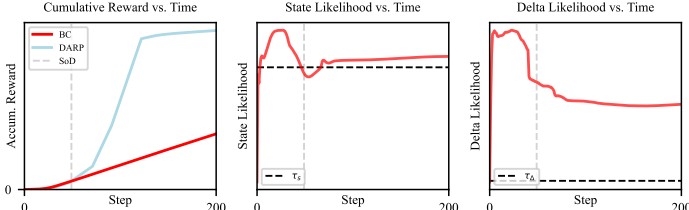

Figure 4: **Cumulative rewards for BC and DARP on the Robosuite stack task illustrate initially identical rollouts that diverge as BC fails the task and DARP succeeds.** A vertical dashed line indicates the step in which the two diverge, labeled "SoD". At the SoD, the state likelihood is $< \tau_s$ (OOD), but the delta likelihood is $> \tau_\Delta$ (in distribution).

## 4 RELATED WORK

**Non-Parametric Imitation Learning Methods:** Non-parametric IL algorithms demonstrate surprising performance by leveraging local structure. VINN Pari et al. (2022) explores locally weighted regression for imitation, showing surprising results in image embedding spaces, and MiDiGaP von Hartz et al. (2025) uses mixtures of Gaussian processes to model multimodal trajectories and achieve rapid generalization. SEABO Lyu et al. (2024) uses retrieval methods to perform offline RL by rewarding transitions close to neighbors to form a reward function. FlowRetrieval Lin et al. (2024), STRAP Memmel et al. (2025), and Behavior Retrieval Du et al. (2023) perform non-parametric retrieval and finetuning from large unlabeled datasets, enabling generalization through test-time training. DARP differs from the above in its unique parameterization of retrieved states into $(s_i^*, a_i^*, s_i^* - s_q)$ tuples and learning a semi-parametric policy rather than relying purely on non-parametric aggregation or test-time training. This provides the variance reduction of local methods and generalization of parametric policies.

**Smoothness-Constrained Policy Learning:** Much recent literature has explored explicit smoothness constraints to improve policy stability and robustness. L2C2 Kobayashi (2022) considers model-free RL under local Lipschitz continuity constraints, achieving smoothness and noise robustness without sacrificing expressiveness, while Asadi et al. (2018) proposed a similar methodology for model-based RL models with Lipschitz constraints. CCIL Ke et al. (2024a) extends these ideas to generate synthetic corrective labels for imitation learning using a Lipschitz-constrained dynamics model. This has also been scaled up to humanoid controllers Chen et al. (2024) to reduce shakiness on deployment. DARP differs from these methods by enforcing smoothness implicitly through an architecture change while using standard imitation learning objectives.

**In-Context Learning Methods:** Recent work has explored non-parametric retrieval from the perspective of in-context imitation learning. REGENT Sridhar et al. (2025) investigates retrieval-augmented generalization by incorporating retrieved states, actions, and rewards into a causal transformer, while DPT Lee et al. (2023) uses supervised pretraining for transformers to predict actions given query states and in-context datasets, effectively learning how to explore. Other in-context architectures include ICRT Fu et al. (2024), Instant Policy Vosylius & Johns (2025), and KAT Di Palo & Johns (2024). These methods aim to quickly adapt to new tasks and environments, whereas DARP focuses on accomplishing higher performance and stability on standard imitation learning.

## 5 CONCLUSION

We introduced **D**ifference-**A**ware **R**etrieval **P**olicies for Imitation Learning (**DARP**) (DARP), a nearest-neighbor-based algorithm that reparameterizes the imitation learning problem in terms of relative differences between query states and their nearest neighbors, rather than learning direct state-to-action mappings. We prove that our method implicitly achieves Laplacian smoothing. Our experimental evaluation across diverse domains, including continuous control and robotic manipulation, validates three key hypotheses. First, DARP consistently outperforms standard behavior cloning when using low-dimensional state representation. Second, DARP maintains performance across different state representations, action distribution modeling requirements, and task complexities, with improvements ranging from 15-46% across tested scenarios. Third, architectural ablations reveal that distance vectors and permutation-invariant aggregation are crucial components to our algorithm.

## 6 REPRODUCIBILITY STATEMENT

A link to supplementary source code is provided. This codebase contains all code used to train and evaluate our models. It also contains policy and environment configuration files to generate all results seen in this paper. We provide all data used in MuJoCo experiments and provide scripts to generate expert demonstrations for Robosuite tasks via MimicGen. We also provide all code necessary to transform between different modalities, such as low-dimensional state representation to images to R3M features. Results will be identical to those in the paper on NVIDIA L40 and L40s GPUs, with the exception of results that require the use of a transformer (REGENT, Set Transformer), which are non-deterministic and may differ slightly from reported numbers.

## 7 ACKNOWLEDGMENTS

This work was (partially) funded by grants from the National Science Foundation NRI (#2132848), DARPA RACER (#HR0011-21-C-0171), and the Office of Naval Research (#N00014-24-S-B001 and #2022-016-01 UW). We gratefully acknowledge gifts from Amazon, Collaborative Robotics, Cruise, the Toyota Research Institute under the University 2.0 program, the Research Scholarship from the Mary Gates Endowment for Students, and others.

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

# A APPENDIX

## A.1 LEMMAS AND PROOFS

### A.1.1 PROOF OF THEOREM 1

To prove Theorem 1, we first start with a well-known result (Chung, 1997; Zhou et al., 2003; Belkin & Niyogi, 2008).

**Lemma 1** (Smoothness regularizer as $k$-NN graph Laplacian penalty). *Let* $\{s_1, \ldots, s_n\}$ *be the expert states with corresponding predicted actions* $f(s_i) \in \mathbb{R}^{d_a}$. *For each i, let* $\mathcal{N}_k(s_i)$ *denote the indices of the k-nearest neighbors of* $s_i$ *(excluding i). Define asymmetric weights*

$$\tilde{W}_{ij} = \begin{cases} w_j(s_i), & \text{if } j \in \mathcal{N}_k(s_i), \\ 0, & \text{otherwise}, \end{cases}$$

*and construct a symmetric affinity matrix*

$$W_{ij} = \tfrac{1}{2}\big(\tilde{W}_{ij} + \tilde{W}_{ji}\big).$$

*Let D be the degree matrix with* $D_{ii} = \sum_j W_{ij}$, *and define the k-NN graph Laplacian* $L = D - W$.

*Then the smoothness regularizer can be written as the quadratic form*

$$\mathcal{L}_S(f) = \frac{1}{n} \sum_{i=1}^{n} \sum_{j \in \mathcal{N}_k(s_i)} w_j(s_i) \|f(s_i) - f(s_j)\|^2 \propto \text{Tr}\big(F^\top L F\big),$$

*where* $F = [f(s_1), f(s_2), \ldots, f(s_n)]^\top \in \mathbb{R}^{n \times d_a}$. *Equivalently, in the scalar case,*

$$\mathcal{L}_S(f) \propto f^\top L f.$$

**Corollary 1** (Continuum limit of smoothness regularizer). *Assume states* $\{s_i\}_{i=1}^{n}$ *are sampled i.i.d. from a smooth density* $p(s)$ *supported on an m-dimensional* $C^2$ *manifold* $\mathcal{M} \subset \mathbb{R}^d$. *Let W be the symmetrized k-NN affinity matrix constructed from a kernel* $K_\Delta$ *with bandwidth h, and let* $L = D - W$ *be the graph Laplacian.*

*If* $n \to \infty$, $h \to 0$, *and* $nh^{m+2} \to \infty$, *then the normalized quadratic form converges to the weighted Dirichlet energy:*

$$\frac{1}{n^2 h^{m+2}} \text{Tr}\big(F^\top L F\big) \longrightarrow C_K \int_{\mathcal{M}} \|\nabla_{\mathcal{M}} f(s)\|_2^2 \, p(s)^2 \, d\text{vol}(s),$$

*where* $C_K > 0$ *is a constant depending only on the kernel* $K_\Delta$.

**Theorem 1** (Manifold Regularized BC ($\mathcal{L}_{\text{MRIL}}$) improves over vanilla BC ($\mathcal{L}_{\text{BC}}$)). *Let* $f^* : \mathscr{S} \to \mathscr{A}$ *be the true, underlying expert policy, assumed to be* $C^2$*-smooth on a compact state space* $\mathscr{S}$. *Let* $f : \mathscr{S} \to \mathscr{A}$ *denote the learned policy estimator. Consider two estimators trained on expert demonstrations:*

1. ***Vanilla BC:*** *a global supervised model minimizing*

$$\mathcal{L}_{\text{BC}}(f) = \mathbb{E}_{(s,a) \sim \mathscr{D}^*}[\ell(f(s), a)].$$

2. ***MRIL:*** *a neighbor-based estimator minimizing*

$$\mathcal{L}_{\text{MRIL}}(f) = \mathcal{L}_{\text{BC}}(f) + \lambda \mathbb{E}_{s \sim \mathscr{D}^*}\left[\sum_{i \in \mathcal{N}_k(s)} w_i(s) \big\|f(s) - f(s_i^*)\big\|_2^2\right],$$

*where* $w_i(s)$ *are the kernel weights defined above and* $\lambda > 0$.

*Then, under the smoothness assumption on f, the following hold:*

*(i)* Variance reduction: *The Laplacian penalty in MRIL acts as a data-dependent Tikhonov regularizer, yielding smaller estimator variance than vanilla BC.*

    *(ii)* Smoothness guarantee: *Minimizers of $\mathscr{L}_{\mathrm{MRIL}}$ satisfy a uniform bound on the local Lipschitz constant of $f$, whereas vanilla BC admits interpolants with arbitrarily large Lipschitz constants between training states.*

    *(iii)* Policy stability: *In a closed-loop rollout, the deviation recursion*

$$\Delta_{t+1} \leq L_s\Delta_t + L_a\|\pi(s_t) - f(s_t^*)\|^4$$

*accumulates error linearly for vanilla BC, but sublinearly for MRIL, since the smoothness regularizer enforces $\|f(s) - f(s')\| = O(\|s - s'\|)$ for neighbors $s, s'$.*

*This suggests that MRIL enjoys strictly better generalization and stability guarantees than BC.*

*Proof.* We prove each claim in turn.

**(i) Variance reduction.** The Laplacian penalty in $\mathscr{L}_{\mathrm{MRIL}}$ is

$$\sum_{i,j} W_{ij}\|f(s_i) - f(s_j)\|^2 = 2f^\top Lf,$$

where $L = D - W$ is the graph Laplacian, and $W$ is the $k$-NN affinity matrix. By Lemma 1, this equals the empirical Dirichlet energy of $f$ on the $k$-NN graph. It is well known (Zhou et al., 2003; Belkin & Niyogi, 2008) that such a quadratic penalty is equivalent to Tikhonov regularization with respect to the graph Laplacian norm $\|f\|_L^2 = f^\top Lf$. In statistical learning theory, adding a Tikhonov penalty strictly reduces the variance of the estimator compared to the unregularized solution while keeping the bias term controlled. Thus MRIL enjoys smaller estimator variance than vanilla BC, which uses no such penalty.

**(ii) Smoothness guarantee.** Consider the continuum limit (Corollary 1): for i.i.d. samples $\{s_i\}$ from density $p$ on a smooth manifold $\mathscr{M}$, the normalized penalty converges to

$$\int_{\mathscr{M}} \|\nabla f(s)\|^2 p(s)^2 \, d\mathrm{vol}(s).$$

This is the weighted Dirichlet energy of $f$ on $\mathscr{M}$. If this integral is finite, $f$ belongs to the Sobolev space $H^1(\mathscr{M}, p^2)$, and in particular $f$ is locally Lipschitz almost everywhere with

$$\|f(s) - f(s')\| \leq C\|s - s'\| \quad \text{for } p\text{-a.e. neighbor pairs } s, s'.$$

Therefore minimizers of $\mathscr{L}_{\mathrm{MRIL}}$ have uniformly bounded local Lipschitz constants along high-density regions of the state space. By contrast, minimizers of vanilla BC have no such constraint: any oscillatory interpolant that matches the training data exactly yields the same supervised risk, so arbitrarily large Lipschitz constants are possible.

**(iii) Policy stability.** Let $\Delta_t = \|s_t - s_t^*\|$ denote the deviation at time $t$. For Lipschitz dynamics $T$,

$$\Delta_{t+1} \leq L_s\Delta_t + L_a\|\pi(s_t) - f(s_t^*)\|.$$

Decompose the action error:

$$\|\pi(s_t) - f(s_t^*)\| \leq \|\pi(s_t) - f(s_t)\| + \|f(s_t) - f(s_t^*)\|.$$

For vanilla BC, the first term $\|\pi(s_t) - f(s_t)\|$ is only minimized on the empirical distribution $P_{\mathscr{S}}$; off-distribution, it may be $O(1)$ regardless of $\Delta_t$. The second term satisfies $\|f(s_t) - f(s_t^*)\| = O(\Delta_t)$ by smoothness of $f$. Hence the recursion can take the form

$$\Delta_{t+1} \leq L_s\Delta_t + L_a\big(O(1) + O(\Delta_t)\big),$$

which accumulates linearly in $t$.

For MRIL, the Laplacian penalty enforces

$$\|\pi(s) - \pi(s')\| \leq C\|s - s'\| \quad \text{for neighbor pairs } (s, s'),$$

---

[4]$L_s$ and $L_a$ are the Lipschitz constants of the environment transition dynamics with respect to state and action, respectively. We denote $\pi(s_t)$ as the action predicted by the agent from the state at time $t$, $s_t$.

as shown in part (ii). Thus $\|\pi(s_t) - f(s_t)\| = O(h^2)$ (where $h$ is the radius of the kernel bandwidth around the point $s_t$) by local-linear regression error bounds, and $\|f(s_t) - f(s_t^*)\| = O(\Delta_t)$ by smoothness of $f$. Combining these,

$$\Delta_{t+1} \leq L_s \Delta_t + L_a \big( O(h^2) + O(\Delta_t) \big).$$

Since the constant multiplying $\Delta_t$ is strictly smaller under the smoothness constraint, the cumulative error grows strictly slower than in vanilla BC. In particular, error growth is sublinear in the rollout horizon when $h$ is small, whereas it is linear for vanilla BC.

**Conclusion.** Claims (i)–(iii) establish that MRIL yields lower variance, uniform smoothness control, and sublinear rollout error accumulation compared to vanilla BC, completing the proof. $\square$

**Kernel choice.** For the IC smoothness regularizer, we adopt a Gaussian kernel

$$w_i(s) \propto \exp\Big( -\frac{\|s - s_i^*\|^2}{2h^2} \Big), \qquad \sum_{i \in \mathcal{N}_k(s)} w_i(s) = 1,$$

with bandwidth $h$ set to the median distance to the $k$-th nearest neighbor across the dataset. This choice is standard in manifold regularization (Belkin & Niyogi, 2008; Zhou et al., 2003) and ensures that the graph Laplacian penalty converges to the Dirichlet energy in the continuum limit. In practice, we found this default to be stable across tasks, though other kernels (e.g., uniform $k$-NN or exponential decay) yield qualitatively similar results.

### A.1.2 PROOF OF THEOREM 2

We begin with the required Lemmas establishing the spectral form of explicit Laplacian regularization and neighbor aggregation.

**Lemma 2** (Spectral form of explicit Laplacian regularization). *Let L be the symmetric normalized graph Laplacian with eigenpairs $\{(\mu_j, u_j)\}_{j=1}^n$, where $0 = \mu_1 \leq \mu_2 \leq \cdots \leq \mu_n \leq 2$. The minimizer of the penalized objective*

$$\mathcal{L}_\lambda(f) = \|f - a^*\|^2 + \lambda f^\top L f$$

*has the closed-form expansion*

$$f_\lambda = \sum_{j=1}^n \frac{1}{1 + \lambda \mu_j} \langle a^*, u_j \rangle u_j.$$

*Thus $\lambda$ directly determines the spectral filter $\phi_\lambda(\mu) = (1 + \lambda\mu)^{-1}$ applied to each Laplacian mode.*

*Proof.* Diagonalize $L = U\Lambda U^\top$ with $U = [u_1, \dots, u_n]$ orthogonal and $\Lambda = \mathrm{diag}(\mu_1, \dots, \mu_n)$. Write $f = Uc$, $a^* = Ub$ in this basis. The objective becomes

$$\|Uc - Ub\|^2 + \lambda c^\top \Lambda c = \sum_{j=1}^n (c_j - b_j)^2 + \lambda \mu_j c_j^2.$$

Minimizing each term yields $c_j = \frac{1}{1 + \lambda\mu_j} b_j$. Transforming back gives the stated expansion. $\square$

**Lemma 3** (Spectral form of neighbor aggregation). *Let $S = D^{-1}A$ be the random-walk matrix of the $k$-NN graph, with adjacency $A$ and degree $D$. For any prediction vector $f$, the neighbor-averaged prediction is $\hat{f} = Sf$. In the Laplacian eigenbasis, this corresponds to the spectral filter*

$$\hat{f} = \sum_{j=1}^n (1 - \mu_j) \langle f, u_j \rangle u_j,$$

*i.e. $\phi_{\mathrm{DARP}}(\mu) = 1 - \mu$.*

*Proof.* By definition, $L = I - D^{-1/2}AD^{-1/2}$ and $S = D^{-1}A = I - L_{\mathrm{rw}}$ where $L_{\mathrm{rw}} = D^{-1}L$ is the random-walk Laplacian. Since $L_{\mathrm{rw}}$ and $L$ share the same spectrum up to similarity transform, the eigenbasis $\{u_j\}$ diagonalizes $S$. Thus for each mode $u_j$, $Su_j = (1 - \mu_j)u_j$, yielding the claimed spectral filter. $\square$

**Theorem 2** (iMRIL is parameter-free Laplacian regularization for BC (MRIL)). *Consider the symmetric normalized k-NN graph Laplacian L (defined in Section 2.2), with eigenpairs $\{(\mu_j, u_j)\}_{j=1}^n$, where $0 = \mu_1 \leq \mu_2 \leq \cdots \leq \mu_n \leq 2$.*

*The minimizers of the explicit MRIL objective (Section 2.2) and the implicit iMRIL objective (Section 2.3) have the following closed form expansions*

$$f_{\text{MRIL}} = \sum_{j=1}^n \frac{1}{1 + \lambda \mu_j} \langle a^*, u_j \rangle u_j \qquad \hat{f}_{\text{iMRIL}} = \sum_{j=1}^n (1 - \mu_j) \langle f, u_j \rangle u_j$$

*iMRIL 's neighbor aggregation step applies the fixed spectral filter $\phi_{\text{iMRIL}}(\mu) = 1 - \mu$ to the graph Laplacian L, preserving low-frequency modes and suppressing high-frequency modes. The congruence between $\hat{f}_{\text{iMRIL}}$ and $f_{\text{MRIL}}$ shows that iMRIL is equivalent to a built-in form of Laplacian smoothing (MRIL) with effective $\lambda \approx 1$ in normalized units. Unlike explicit regularization, this implicit filter requires no additional hyperparameter tuning.*

*Proof.* From Lemma 3, the neighbor aggregation operator $S = D^{-1}A$ acts on Laplacian eigenmodes $u_j$ as

$$S u_j = (1 - \mu_j) u_j,$$

where $\mu_j$ are the normalized Laplacian eigenvalues. Thus in the graph Fourier basis, neighbor aggregation corresponds to multiplying each mode by the fixed spectral filter $\phi_{\text{DARP}}(\mu) = 1 - \mu$.

On the other hand, Lemma 2 shows that explicit Laplacian regularization with parameter $\lambda$ yields the spectral filter $\phi_\lambda(\mu) = (1 + \lambda \mu)^{-1}$. Both filters downweight high-frequency modes ($\mu \gg 0$) while preserving low-frequency modes ($\mu \approx 0$). The key difference is that $\phi_\lambda(\mu)$ requires tuning $\lambda$, whereas $\phi_{\text{DARP}}(\mu)$ is parameter-free.

To see the equivalence, note that for small $\mu$,

$$\phi_{\text{DARP}}(\mu) = 1 - \mu \approx (1 + \mu)^{-1} = \phi_{\lambda=1}(\mu) \quad \text{up to } O(\mu^2) \text{ terms.}$$

Thus DARP can be interpreted as performing Laplacian smoothing with an effective regularization weight of order $\lambda \approx 1$ in normalized units. Moreover, for large $\mu$, $\phi_{\text{DARP}}(\mu)$ damps high-frequency modes even more strongly by driving them toward zero, providing a sharper low-pass effect than explicit regularization.

Therefore, DARP 's aggregation step is mathematically equivalent to implicit Laplacian regularization with fixed spectral filter $\phi_{\text{DARP}}$, eliminating the need to tune $\lambda$ explicitly. $\square$

DARP can therefore be viewed as a form of *locally adaptive implicit regularization*: rather than introducing an explicit global weight $\lambda$, its neighbor aggregation step enforces smoothness automatically through the graph structure. The effective regularization strength varies with local degree

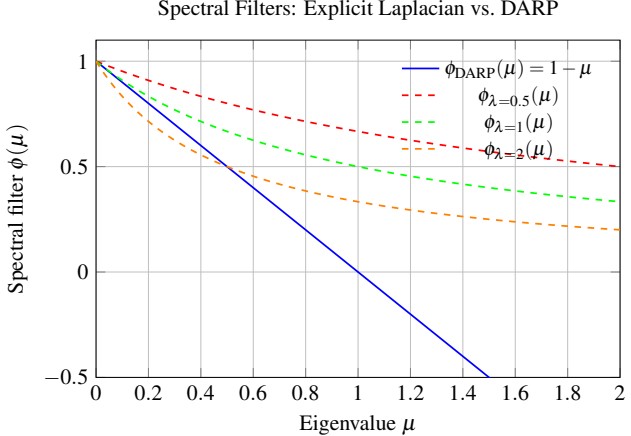

Figure 5: DARP achieves sharper low-pass filtering

and neighborhood geometry, adapting to the density of the expert demonstrations. Spectrally, this corresponds to the fixed filter $\phi_{\text{DARP}}(\mu) = 1 - \mu$, which suppresses high-frequency modes more aggressively than any fixed explicit $\lambda$. Figure 5 illustrates this comparison, showing how DARP achieves sharper low-pass filtering without the need for hyperparameter tuning.

## A.2 ADDITIONAL EXPERIMENTAL DETAILS

### A.2.1 RETRIEVAL

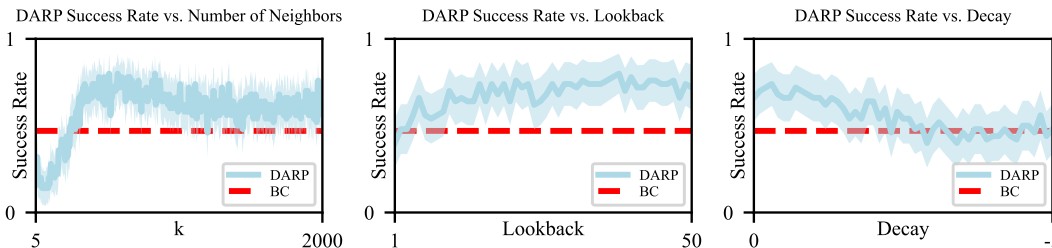

Figure 6: DARP performance analysis as retrieval hyperparameters are swept: (left) observe that the performance of a DARP model is poor when using few neighbors, reaches a global optimum when retrieving about 500 neighbors, and plateaus just above BC's success rate as $k$ goes to the size of the dataset; (center) observe that the performance of a DARP model generally slightly improves as more history is considered, and only performs worse than BC when very little or no history is considered; (right) observe that the performance of a DARP model is sensitive to how much weight is applied to older observations when performing retrieval. Intuitively, if this decay is too high, DARP performance is nearly identical to having little to no lookback, performing worse than BC. Success rate is measured out of 50 trials on the Robosuite Stack environment. 95% confidence intervals are included.

The selection of the distance function $d(s_q, s_i^*)$ to select $k$ neighbors is crucial. While we find that simple Euclidean distance between states can work, in our experiments, we use a slightly modified algorithm that takes advantage of the fact that we are working with sequences of states and incorporates history in our distance calculation.

Suppose we have a query trajectory $S_q = (\ldots, s_{q,-1}, s_{q,0})$ where $s_{q,0}$ is the current query state $s_q$. Now suppose we want to calculate $d(s_q, s_i^*)$, where $s_i^*$ is some state from the expert dataset. We first find the trajectory this state is from—call this $S_j^*$—and the index of $s_i^*$ in this trajectory—call this $t$. Thus, $s_i^*$ can be rewritten as $s_{j,t}^*$. Given some lookback parameter $\ell$ which denotes how many past states we want to consider, we get:

$$d(s_q, s_i^*) = \sum_{n=0}^{\ell-1} \|s_{q,-n} - s_{j,t-n}^*\|$$

This is simply the accumulation of Euclidean distances of the current and last $\ell - 1$ states from both the query trajectory and the source trajectory, assuming valid indices. Of course, in practice, we generally want to put more emphasis on more recent states, as we want them to be more influential in the selection of neighbors. Thus, given some rate of exponential decay $\gamma \geq 0$, we have

$$d(s_q, s_i^*) = \sum_{n=0}^{\ell-1} \|s_{q,-n} - s_{j,t-n}^*\| \cdot e^{-\gamma n}$$

See Figure 6 for an experimental analysis on how the success rate in an environment changes as these parameters are swept.

### A.2.2 CAN DARP HANDLE TASKS REQUIRING THE REPRESENTATION OF MULTI-MODAL ACTION DISTRIBUTIONS?

We test DARP's ability to handle complex action distributions by evaluating on the Push-T task, as described in (Chi et al., 2024), which requires representing multi-modal action distributions (see Figure 7 for a visualization). For this experiment, DARP employs a Set Transformer head that predicts parameters of a Gaussian Mixture Model. We note that DARP with a GMM head is able to handle multi-modal distributions effectively, showing a 22% improvement over BC on the Push-T

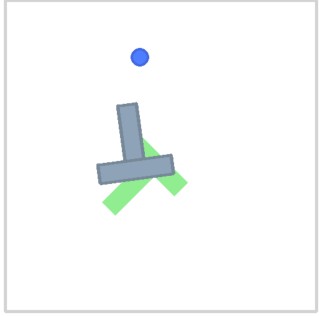

| Method | Score |
|---:|:---|
| BC | 48 ± 8 |
| **DARP** | **70 ± 8** |

Table 5: **Push-T Results.** Averaged over 100 trials, DARP outperforms BC.

Figure 7: **Push-T Environment.** The goal is to control the blue circle to push the T-shaped block.

task (Q2) (see Table 5). This demonstrates that DARP can be further adapted to multi-modal action distribution modeling requirements.

### A.2.3 CAN DARP HANDLE DISCONTINUOUS ENVIRONMENTS WHERE NEARBY STATES MAY REQUIRE OPPOSING ACTIONS?

A key concern for neighbor-based approaches is performance in environments with strong discontinuities, where states that are close in Euclidean distance may require drastically different actions. To address this concern, we design a stress test using a modified version of D4RL's Umaze environment (see Figure 8 for a visualization).

**Even in this deliberately challenging discontinuous environment, DARP achieves a 57% success rate compared to BC's 25%. (Q3) (see Table 6)** This suggests that the distance vectors and permutation-invariant aggregation help the model distinguish between appropriate and inappropriate neighbors, even when spatial proximity doesn't guarantee action similarity.

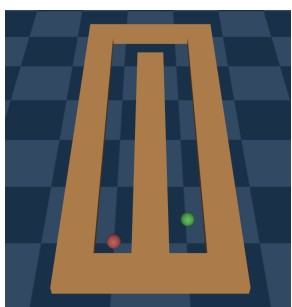

| Method | Succ. (%) |
|---:|:---|
| BC | 25 |
| **DARP** | **57** |

Table 6: **Long maze results.** Averaged over 100 trials, DARP significantly outperforms BC.

Figure 8: **Long maze environment.** The goal is to move a force-actuated ball from the green start to the red destination.

### A.2.4 CAN DARP RECOVER FROM BC ERROR?

In order to analyze DARP's robustness to accumulated error, we roll out a BC agent in an environment in which we know it will fail, but every $k$ steps, we create a fork of the environment and begin rolling out a DARP agent in that clone of the environment. The results (seen in Fig. 9) show that, even as BC approaches failure and drifts away from the support of expert demonstrations, DARP is able to recover and score very highly. This suggests that DARP indeed has superior robustness to accumulation of error and can perform well in the slightly out-of-distribution states that a failing BC agent drifts into.

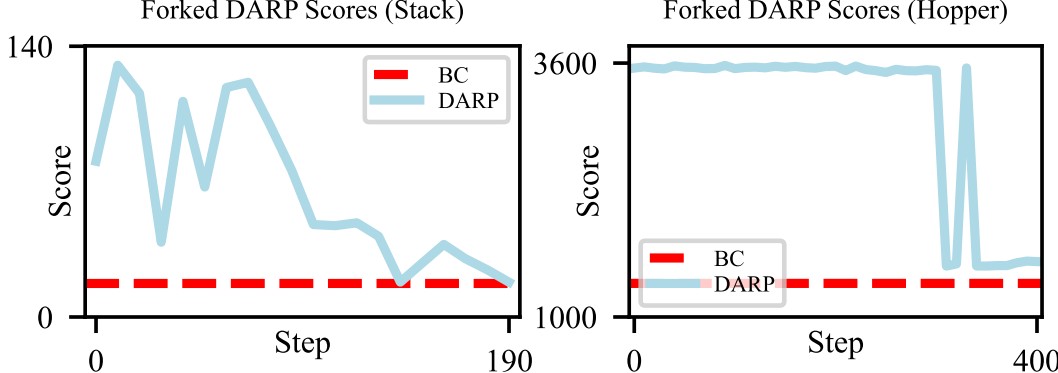

Figure 9: In two different tasks (the Robosuite Stack task and the MuJoCo Hopper task), we rollout a BC agent and create a fork of the environment every $k$ steps (in this case, $k = 10$). Observe that, even as BC nears the end of its failing rollout, DARP is able to scale highly, and is only prevented from doing so about halfway through the Stack rollout and about 80% through the Hopper rollout.

| Environment | Method | Improvement over BC (%) |
|---|---|---|
| **Hopper** | CCIL | 32.6% |
| | **DARP** | **53.2%** |
| **Walker** | **CCIL** | **84.1%** |
| | **DARP** | **84.1%** |
| **Ant** | CCIL | 12.6% |
| | **DARP** | **84.5%** |
| **HalfCheetah** | CCIL | 5.4% |
| | **DARP** | **418.7%** |

Table 7: **Relative improvement over BC compared to CCIL.** Comparing DARP against CCIL on standard MuJoCo benchmarks. DARP achieves higher or equal relative gains across all tasks.

### A.2.5 COMPARISON WITH CCIL

As shown in Table 7, we evaluate our method against CCIL, a baseline that explicitly induces smoothness. We use the reported scores in the CCIL paper, and compare percent improvement over BC. Observe that DARP outperforms CCIL significantly on three out of four environments, with a particularly large margin on HalfCheetah (418.7% vs 5.4% improvement).

### A.2.6 DISTANCE METRIC SENSITIVITY

| Method | Success Rate (%) |
|---|---|
| BC | 47 |
| **DARP (Cosine Similarity)** | 70 |
| **DARP (Euclidean Distance)** | **75** |

Table 8: **Distance Metric Sensitivity.** Comparison of success rates using R3M features. DARP with Euclidean distance and cosine similarity perform similarly, beating BC by 28% and 23% respectively. Results are on the Robosuite Stack task.

While all experiments performed in this paper use Euclidean distance to choose nearby neighbors, it is natural to consider alternative metrics, like cosine similarity, especially in high-dimensional embeddings such as R3M. We find (as shown in Table 8) that DARP performs similarly – only 5% worse – when using cosine similarity rather than Euclidean distance. This suggests DARP is robust to the choice of distance metric used.

### A.2.7 DARP in Combination With Diffusion Policy

| Method | Stack | Threading | Peg Insertion |
|---|---|---|---|
| BC (MLP) | 34 | 12 | 28 |
| DARP (MLP) | 62 | 38 | 40 |
| Diffusion | 52 | 44 | 54 |
| **DARP w/ Diffusion** | **82** | **72** | **66** |

Table 9: **DARP in combination with diffusion policy.** DARP provides significant improvements when applied to both standard MLP policies and Diffusion policies. Results are on the Robosuite Stack, Threading, and Peg Insertion tasks as success rate across 50 trials. States are state-based. Note that the number of demonstrations used is much less than that used in Table 3, so baseline BC and DARP numbers do not match.

While all reported experiments are performed with an MLP backbone, diffusion policy (Chi et al., 2024) has proven to be a state-of-the-art model class for imitation learning, particularly for manipulation tasks. Table 9 reveals that DARP is not mutually exclusive with diffusion and can be combined for an even more performant model. Using the DARP architecture with a diffusion backbone outperforms DARP with an MLP backbone by 20%, 34%, and 26% success rate increase for all three tasks respectively, beating standard BC by a total 48%, 60%, and 38% for the three tasks.

| Method | Training (s/epoch) | Inference (s/step) |
|---|---|---|
| Diffusion | 5.610 | 0.15700 |
| **DARP (MLP)** | **0.559** | **0.00437** |

Table 10: **Computational efficiency of DARP with an MLP backbone and diffusion policy.** Comparison of runtime costs between DARP (MLP) and diffusion policy. Diffusion policy is significantly more expensive, being $\approx 10\times$ slower in training and $\approx 36\times$ slower during inference.

Additionally, we empirically find that DARP with an MLP backbone is much faster than standard diffusion, particularly in inference – see Table 10.

### A.2.8 DARP Trained on Human Demonstrations

| Method | Success Rate (%) |
|---|---|
| BC | 45 |
| **DARP** | **60** |

Table 11: **Results with human demonstrations.** Comparison of success rates when training on human data rather than data collected by RL policies. Results are on the Robosuite Stack task.

The expert demonstrations used to train models for the MuJoCo and Robosuite environments are collected by an optimal Reinforcement Learning policy. It is crucial to ensure DARP maintains a performance gain in comparison to BC when trained on expert demonstrations collected by humans. Indeed, when trained on human demonstrations on the Robosuite Stack task, DARP outperforms standard BC by 15% (see Table 11).

### A.2.9 CHOICE OF RETRIEVAL HYPERPARAMETERS

| Environment | Method | Score (Mean) |
|---|---|---|
| **Hopper** | BC | 507.49 |
| | DARP | **805.69** |
| **Stack** | BC | 0.12 |
| | DARP | **0.31** |

Table 12: **Performance comparison when validation loss is used to select training epochs and retrieval hyperparameters.** Mean scores on Hopper and Stack environments. Observe that DARP maintains a performance gain in comparison to BC.

Figure 6 reveals that there are selections of retrieval parameters (for example, a very low number of neighbors) which cause DARP to perform worse than standard BC. However, we find that choosing retrieval hyperparameters that minimize validation loss is an effective strategy to find performant settings, see Table 12.

### A.3 PSEUDOCODE

We provide pseudocode of the DARP algorithm, see Algorithm 1.

---

**Algorithm 1** Difference-Aware Retrieval Policies

---

1: **Input:** Expert demonstrations $\mathscr{D}^* = \{(s_j^*, a_j^*)\}$, number of neighbors $k$
2: **Initialize:** $f$ parameters $\theta$
3: **if** $g$ is parametric **then**
4:     **Initialize:** $g$ parameters $\psi$
5: **end if**
6: // Training Loop
7: **while** not converged **do**
8:     Sample batch of query data $(s_q, a_q) \sim \mathscr{D}^*$
9:     **for** each query pair $(s_q, a_q)$ in batch **do**
10:         // Find $k$-Nearest Neighbors from the entire dataset $\mathscr{D}^*$
11:         $\mathscr{N}_k(s_q) \leftarrow \arg\min\text{-}k_j d(s_q, s_j^*)$
12:         // Compute Neighbor-based Predictions
13:         **for** each neighbor index $i \in \mathscr{N}_k(s_q)$ **do**
14:             $a_i' \leftarrow f_\theta(s_i^*, a_i^*, s_i^* - s_q)$
15:         **end for**
16:         // Aggregate Predictions
17:         **if** $g$ is parametric **then**
18:             $\hat{a}_q \leftarrow g_\psi(\{a_i'\}_{i \in \mathscr{N}_k(s_q)})$
19:         **else**
20:             $\hat{a}_q \leftarrow g(\{a_i'\}_{i \in \mathscr{N}_k(s_q)})$
21:         **end if**
22:     **end for**
23:     // Update Parameters based on the batch loss
24:     $\mathscr{L} \leftarrow \sum_{(s_q, a_q) \in \text{batch}} \|\hat{a}_q - a_q\|^2$
25:     // Gradient descent step
26:     $\theta \leftarrow \theta - \alpha \nabla_\theta \mathscr{L}$
27:     **if** $g$ is parametric **then**
28:         $\psi \leftarrow \psi - \alpha \nabla_\psi \mathscr{L}$
29:     **end if**
30: **end while**
31: **Output:** Trained parameters $\theta$ and, if applicable, $\psi$

---

## A.4 RUNTIME ANALYSIS

| Environment | $k$ | Train (s/epoch) | Test (s/step) |
|---|---|---|---|
| **Hopper (1,000 datapoints)** | BC | 0.102 | 0.00127 |
| | 100 | 0.126 | 0.00413 |
| | 200 | 0.128 | 0.00418 |
| | 300 | 0.130 | 0.00420 |
| | 400 | 0.131 | 0.00421 |
| | 500 | 0.130 | 0.00419 |
| **Stack (4,200 datapoints)** | BC | 0.431 | 0.00139 |
| | 100 | 0.556 | 0.00437 |
| | 250 | 0.539 | 0.00442 |
| | 500 | 0.559 | 0.00437 |
| | 750 | 0.576 | 0.00433 |
| | 1000 | 0.591 | 0.00478 |
| | 1500 | 0.682 | 0.00452 |
| | 2000 | 0.758 | 0.00451 |

Table 13: **Runtime comparison across environments.** Training and testing speeds (in seconds) for Behavior Cloning (BC) and varying values of $k$ on Hopper and Stack datasets.

As shown in Table 13, we analyze the computational overhead of DARP at both training-time and inference-time. Our analysis indicates that computation cost scales sub-linearly as $k$ increases, and that inference time is tractable for real-time robotic application. For example, on the Robosuite Stack task at $k = 500$, inference takes approximately 0.00437 seconds per step on our hardware, corresponding to a control frequency higher than 230 Hz.

