# OpenReview forum: "Difference-Aware Retrieval Policies for Imitation Learning"
_ICLR.cc/2026/Conference — ICLR 2026 Poster_

### Official Review · Reviewer_iFBA · 2025-10-30

**Soundness:** 4
**Presentation:** 3
**Contribution:** 3
**Rating:** 6
**Confidence:** 5

**Summary:**

Authors propose a way to combine retrieval based IL with standard BC that will turns out to be equivalent to manifold smoothed BC. Authors add both theoretical analysis and experimental validation in terms of standard MuJoCo IL experiments and ablation studies.

**Strengths:**

- Interesting, theoretically motivated and practical offline IL method.
- MuJoCo and robosuite experimental results are quite good.
- Narrative is well written and easy to follow.

**Weaknesses:**

- Even though narrative is well written, the mathematical exposition leaves room for improvement. For example looking at page 3, we see a lot of symbols that have not been defined: s^*, \pi(s_t), L, ...
- Even though I understand that focus is in offline IL, it would be good obtain results from IL methods that have access to environment, such as adversarial methods (GAIL, AIRL etc). Can the proposed method bridge the gap between BC and GAIL?
- It appears that expert demos have been obtained from an optimal RL policy, or have I been mistaken? In the "What Matters for Adversarial Imitation Learning?" -paper authors note that there is a difference when using expert demos from optimal policy and expert demos from human expert. It would be good to test with such demos, one option is to use ALE environment and with Atari-HEAD expert demo dataset.

**Questions:**

- How would you contrast the proposed method to the retrieval IL method presented in: Federico Malato, Ville Hautamäki, "Online Adaptation for Enhancing Imitation Learning Policies", CoG 2024?
- In Section 2.4.2 you show an interesting way to go beyond simple averaging, but in experiments I see no improvement in using it. Why is that? I would suggest not to bold DARP Set Transformer numbers in Table 1 as all of those numbers are worse than plain DARP.

---

> ### Author Response · Authors · 2025-11-25
>
> **“there is a difference when using expert demos from optimal policy and expert demos from human expert. It would be good to test with such demos”**
>
> You are correct that expert demonstrations have been obtained through an RL policy. Mimicgen [1], the algorithm we use to generate our expert demonstrations for Robosuite environments, provides a small amount of human demonstrations on each task. Training on those, we found that BC had a 45% success rate, while DARP had a 60% success rate, demonstrating that DARP gains over BC persist even on human demonstrations.
>
> **"How would you contrast the proposed method to the retrieval IL method presented in: Federico Malato, Ville Hautamäki, "Online Adaptation for Enhancing Imitation Learning Policies", CoG 2024?"**
>
> Reading "Online Adaptation for Enhancing Imitation Learning Policies,” there are certainly similarities in that actions at inference-time are influenced by retrieved nearby neighbors from the expert dataset. However, the mentioned paper performs online Bayesian adaptation, while our work proposes an offline architectural change in which weights are frozen at inference-time.
>
> **“In Section 2.4.2 you show an interesting way to go beyond simple averaging, but in experiments I see no improvement in using it. Why is that?”**
>
> You are correct in that using a set transformer did not give us any benefit beyond simple averaging on the MuJoCo and RoboSuite tasks. This is because the benefit of using a set transformer, particularly with a GMM head, is that it provides the DARP architecture with the ability to sample actions from a multi-modal action distribution, which is impossible with simple averaging. This becomes very clear in the Push-T environment, which requires the modeling of a multi-modal action distribution. DARP with simple averaging scores about 34%, while DARP with a Set Transformer and GMM head scores 70%.
>
> **“results from IL methods that have access to environment, such as adversarial methods (GAIL, AIRL etc)”**
>
> Conceptually, methods like GAIL should still have an advantage since they have online sample access. However, they are often difficult to stabilize and scale. We are in the process of running experiments with these methods and will report results shortly.
>
> **“looking at page 3, we see a lot of symbols that have not been defined**
>
> Thank you for the pointers! We have updated these definitions in Sections 2.1 and 2.2, please let us know if they still remain unclear.
>
> [1] Ajay Mandlekar, Soroush Nasiriany, Bowen Wen, Iretiayo Akinola, Yashraj Narang, Linxi Fan, Yuke Zhu, and Dieter Fox. Mimicgen: A data generation system for scalable robot learning using human demonstrations. In the 7th Annual Conference on Robot Learning, 2023.

---

### Official Review · Reviewer_kF51 · 2025-10-31

**Soundness:** 3
**Presentation:** 4
**Contribution:** 3
**Rating:** 6
**Confidence:** 5

**Summary:**

This paper proposes DARP, a new semi-parametric BC method that integrates non-parametric retrieval with parametric prediction. DARP retrieves a set of nearest-neighbor expert demonstrations for a given query state and conditions the policy’s action prediction on both the neighbor states and their difference vectors relative to the query. The retrieved neighbors are aggregated in a permutation-invariant manner, enabling the model to enforce implicit local smoothness without explicit regularization. The authors show theoretically that DARP approximates a Laplacian smoothing operation over the expert k-NN graph, promoting manifold-consistent policies. Empirical results demonstrate the improved stability, generalization, and robustness across various continuous-control tasks.

**Strengths:**

1. The core idea of reparameterizing the policy in terms of relative differences to known expert state-actions is highly intuitive. It anchors the policy's predictions to the ground-truth data manifold, providing a strong and data-centric form of 'regularization'.

1. The authors provide a strong theoretical motivation for their architectural choice. The connection drawn between neighbor aggregation (iMRIL) and explicit manifold regularization (MRIL) via spectral graph theory (Theorem 2) offers a compelling, first-principles explanation for why this method should reduce variance and improve stability, linking it to low-pass filtering on the graph Laplacian.

1. The experimental results are good, and the ablation is impressive and sufficient.

**Weaknesses:**

Overall, the proposed method is inspiring and technically sound, and the paper is easy to read. I only have concerns on the inference cost and the distance metric:

1. The method could suffer from large computational/storage overhead at inference time. For every single decision step, the policy must perform a k-NN search over the entire $N$-point expert dataset and then perform $k$ network forward passes. Also, the method requires storing a large expert dataset or its embeddings, which increases memory and storage requirements by orders of magnitude compared to a standalone policy network.

1. The paper assumes that Euclidean distance (or a simple temporal extension with decay) suffices to identify meaningful neighbors, which strong in high-dimensional or structured state spaces, where Euclidean proximity does not necessarily imply functional similarity.

**Questions:**

1.  Have the authors analyzed the trade-off between inference latency and performance?

1. In the proof of Theorem 1 (iii), what does $r$ represent? There seems a loss of definition.

1. Can the method be extended to other distance metrics?

---

> ### Author Response · Authors · 2025-11-25
>
> **"The method could suffer from large computational/storage overhead at inference time."**
>
> Good point! Computation cost of retrieval is something we were extremely intentional about in the development of this work. Here are some numbers on training and inference speed with our method vs. standard BC:
>
> | Environment | $k$ | Test (s/step) |
> | :--- | :--- | :--- |
> | **Hopper (1k datapoints)** | BC | 0.00127 |
> | | 100 | 0.00413 |
> | | 200 | 0.00418 |
> | | 300 | 0.00420 |
> | | 400 | 0.00421 |
> | | 500 | 0.00419 |
> | **Stack (4.2k datapoints)** | BC | 0.00139 |
> | | 100 | 0.00437 |
> | | 250 | 0.00442 |
> | | 500 | 0.00437 |
> | | 750 | 0.00433 |
> | | 1000 | 0.00478 |
> | | 1500 | 0.00452 |
> | | 2000 | 0.00451 |
>
> 1. Inference overhead
> The computational overhead of DARP is manageable at inference-time. For the Stack task, using $k = 500$, on our hardware, inference time with 500 neighbors is about 0.00437 seconds per step, meaning we could run over 230 Hz, which is likely to be sufficient for nearly all real-time robotic applications, demonstrating that cost at inference time should not be prohibitive.
>
> 2. Scaling with dataset size and $k$
> Empirically, we find that inference-time computation cost scales sub-linearly as $k$ increases. For instance, on the Stack task, increasing $k$ from 500 to 2000 increases inference time by only 3%. As for dataset size scaling, going from Hopper (1000 datapoints) to Stack (4200 datapoints) and keeping $k$ fixed at 500, inference time increases by only 4%. As for significantly larger datasets (those much larger than the ones in this paper), Approximate Nearest Neighbor such as FAISS [1, 2] can be leveraged for efficient retrieval.
>
> We do, however, acknowledge that there is indeed the memory burden of having to store the entire expert dataset at inference-time. However, this is true about nearly all non-parametric methods (see [3, 4, 5]), and thus is unfortunately a necessary cost of our DARP architecture.
>
> **“The paper assumes that Euclidean distance (or a simple temporal extension with decay) suffices to identify meaningful neighbors, which strong in high-dimensional or structured state spaces, where Euclidean proximity does not necessarily imply functional similarity.”**
>
> To address this concern, we have experimented with moving from Euclidean distance to the cosine similarity metric, particularly when using the high-dimensional R3M features. We found that it performs similarly, getting a success rate of 70% versus our reported 75% with Euclidean distance, both handily beating BC’s 47%. However, this is still an open and active area of research for us, and experimenting with learned metrics would be interesting future work.
>
> **“In the proof of Theorem 1 (iii), what does r represent? There seems a loss of definition.**
>
> Thanks for pointing this out. The error term for r stems from local-linear regression, where the approximation error scales as $O(r^2)$ where $r$ is the radius of the ball / kernel bandwidth around the point $s_t$. We have updated this in the paper!
>
> [1] Douze, Matthijs, et al. "The Faiss library.(2024)." arXiv preprint arXiv:2401.08281 (2024).
>
> [2] Johnson, Jeff, Matthijs Douze, and Hervé Jégou. "Billion-scale similarity search with GPUs." IEEE Transactions on Big Data 7.3 (2019): 535-547.
>
> [3] Lewis, Patrick, et al. "Retrieval-augmented generation for knowledge-intensive nlp tasks." Advances in neural information processing systems 33 (2020): 9459-9474.
>
> [4] Du, Maximilian, et al. "Behavior retrieval: Few-shot imitation learning by querying unlabeled datasets." arXiv preprint arXiv:2304.08742 (2023).
>
> [5] Memmel, Marius, et al. "Strap: Robot sub-trajectory retrieval for augmented policy learning." arXiv preprint arXiv:2412.15182 (2024).

---

### Official Review · Reviewer_vWKv · 2025-11-01

**Soundness:** 4
**Presentation:** 3
**Contribution:** 3
**Rating:** 4
**Confidence:** 4

**Summary:**

This paper presents DARP, a method to reduce variances in behavior cloning methods. It conditions the action generation using the retrieved nearest neighbors and aggregates the predictions from individual neighbors to output the final action. The experiments show the improvements over prior methods based on retrieved neighbors and BC on MuJoCo and robosuite tasks.

**Strengths:**

- The paper bridges the non-parametric retrieval-based method and the parametric BC to improve the policy performance.
- The paper provides good analysis, both theoretical and experimental, to show the effect of DARP.

**Weaknesses:**

- The main experiment result is based on MuJoCo tasks, which have a lot more neighbors than manipulation, so it is easier to show the improvements. For manipulation tasks, it is unclear which BC architecture it is employing. If it is an MLP, it is not a well-performing architecture for manipulation; it will need to compare with other BC architectures, like diffusion policy, to show the real improved performance.
- As shown in Appendix A2.1, the performance is significantly affected by the distance measure, e.g.,  the amount of look-back. This can show that the retrieval mechanism outweighs the architectural changes proposed in the paper.
- The motivation of the paper is about improving generalization. However, none of the experiments evaluate the generalization performance of DARP.

**Questions:**

- The step of divergence analysis can be considered as slight evidence that DARP handles OOD better, but how often can we detect the divergence? Also, it seems like this analysis is only possible with deterministic policies. How can we apply the same analysis to the push-T task, where DARP uses a GMM head?

---

> ### Author Response · Authors · 2025-11-25
>
> **“compare with other BC architectures, like diffusion policy, to show the real improved performance.”**
>
> For our BC baselines, we used MLPs. You are correct that MLPs are not as performant as diffusion policies; however, DARP architecture is not mutually exclusive with diffusion techniques. We chose to use MLPs due to their quick training and inference speed, but we are actively working on combining our DARP architecture with diffusion. Empirically, we found that DARP with MLPs still outperformed diffusion, though by a much smaller margin than it did vanilla BC (about 5% on the Stack task). However, DARP with MLPs is much faster (diffusion is about 900% slower in training (0.559 seconds per epoch for DARP versus about 5.61 seconds per epoch for diffusion) and 3600% slower in inference (0.00437 seconds per step of inference for DARP versus 0.157 seconds per step of inference for diffusion)).
>
> Additionally, as you stated, diffusion is particularly well-suited for the manipulation tasks. On the MuJoCo tasks, we found that, in our evaluation, diffusion policies did not outperform even BC with MLPs:
>
> | Environment | Method | Score (Mean) |
> | :--- | :--- | :--- |
> | **Hopper** | Diffusion | 1638 |
> | | BC w/ MLP | 2313 |
> | | DARP | 3546 |
> | **Walker** | Diffusion | 1048 |
> | | BC w/ MLP | 2658 |
> | | DARP | 4894 |
> | **Ant** | Diffusion | 1703 |
> | | BC w/ MLP | 2376 |
> | | DARP | 4383 |
>
> **“the retrieval mechanism outweighs the architectural changes proposed in the paper”**
>
> You are correct in pointing out that there are certainly choices of retrieval hyperparameters that cause our method to do worse than BC. What is required is a performant selection criterion for $k$ that can be applied for model selection, without actually having to run online policy evaluation. While there is no immediately perfect metric, we find that optimizing for a $k$ that minimizes validation loss on a per task basis, can serve as an easy way to avoid tedious hyperparameter tuning. To illustrate this, we report performance on two tasks, Hopper (MuJoCo) and Stack (Robosuite) where retrieval hyperparameters and training epochs are chosen by minimizing validation loss:
>
> | Environment | Method | Score (Mean) |
> | :--- | :--- | :--- |
> | **Hopper** | BC | 507.49 |
> | | DARP | 805.69 |
> | **Stack** | BC | 0.12 |
> | | DARP | 0.31 |
>
> We see that DARP outperforms BC, and actually does so with an even larger margin. On the Hopper task, our reported numbers improve upon BC by 53.2%, while these validation-loss minimizing numbers improve upon BC by 58.8%. For the Stack task, the gap goes from an 89.5% improvement to a 158.3% improvement. So, while the numbers reported in our paper were achieved by performing hyperparameter sweeps and evaluating directly on the environment, using validation loss as a surrogate metric also sees DARP improve upon BC. This becomes important as these methods are scaled up to real world settings, where real-world hyperparameter sweeps over k become expensive if online policy evaluation is needed for each model.
>
> **“none of the experiments evaluate the generalization performance of DARP”**
>
> The generalization of DARP that we demonstrate is not across tasks, but rather within different states for the same task. In particular, where DARP is particularly adept is at avoiding compounding error. So if we examine a trajectory sequentially (as shown in Fig 6), we find that standard behavior cloning policies start to deviate at a point (particularly, deviation in the reward signal from the environment), whereas we find that in those OOD states, DARP policies generalize better, ensuring minimal deviation. This is what helps DARP achieve strong empirical performance on closed loop rollouts.
>
> **“The step of divergence analysis can be considered as slight evidence that DARP handles OOD better, but how often can we detect the divergence?”**
>
> The purpose of our analysis is to scientifically understand the impact of design decisions in DARP, but is not actually required to run these policies. We would like to clarify that this analysis is simply being done post-hoc to understand the method better, and the policies themselves do not require any explicit detection of the divergence.
>
> Additionally, this divergence analysis is performed by just rolling out the agent in the environment. While a DARP architecture with a GMM head is non-deterministic, with careful seeding, the agent will perform identically given identical initial conditions. Thus, we can still compare its reward from the environment to that of a BC agent with the same initial conditions and compare the divergence of the two, and we can do so reproducibly, just as we did for DARP with deterministic heads.

---

> > ### Author Response · Authors · 2025-11-26
> >
> > Regarding our discussion on diffusion policies -- we'd like to follow up with some additional results we've collected on policies that merge DARP architecture with diffusion techniques.
> >
> > On the Robosuite Stack task, we found the following success rates:
> >
> > | Method | Score (Success %) |
> > | :--- | :--- |
> > | BC (MLP) | 34 |
> > | DARP (MLP) | 62 |
> > | Diffusion | 50 |
> > | DARP w/ Diffusion | 76 |
> >
> > These numbers reveal that standard diffusion has a 16% higher success rate compared to standard BC, but combining it with DARP architecture adds an additional 26% to this number, beating standard DARP (with MLPs) by 14%. We're excited about these results, and will continue to explore the combination of these two powerful methods.
> >
> > (Note that the number of expert demonstration we used was about half that used to collect the numbers in the paper on this task, so the numbers for BC and DARP are lower than those reported in the paper on this task).

---

### Official Review · Reviewer_T2hb · 2025-11-07

**Soundness:** 3
**Presentation:** 3
**Contribution:** 3
**Rating:** 6
**Confidence:** 4

**Summary:**

This paper introduces DARP, a retrieval-based imitation learning method that addresses covariate shift in behavior cloning by reparameterizing the problem using local neighborhood structure. The key innovation is conditioning action predictions on k-nearest neighbors along with their actions and crucially, the difference vectors between neighbor states and query states. The authors provide theoretical analysis showing DARP implicitly performs Laplacian smoothing and demonstrate 15-46% performance improvements across continuous control and robotic manipulation tasks.

**Strengths:**

- The use of difference vectors (s*_i - s_q) rather than just neighbor states is creative and well-motivated. The ablation in Figure 5 convincingly shows this is crucial for performance.
- The connection to Laplacian smoothing provides intuition for why the method works and bridges local and global learning paradigms effectively.
- The experiments span MuJoCo locomotion, robotic manipulation, visual observations, and even deliberately discontinuous environments, showing broad applicability.

**Weaknesses:**

-  The paper doesn't discuss the computational cost of k-NN retrieval at every training and inference step. For k=500 (as suggested in Figure 8), this could be prohibitive for large datasets or real-time applications.
- While the paper compares to BC and some retrieval methods, it lacks comparison with other smoothness-inducing approaches mentioned in related work (L2C2, CCIL). The MRIL baseline helps but isn't a published method.
- The jump from iMRIL (which uses simple averaging) to DARP (with difference vectors and learned aggregation) isn't theoretically justified. Does the Laplacian smoothing interpretation still hold?

**Questions:**

- What is the time complexity for training and inference? How does this scale with dataset size and k?
- How sensitive is DARP to the choice of distance metric, especially in high-dimensional spaces? Have you experimented with learned metrics?
-  Does the Laplacian smoothing analysis extend to the full DARP algorithm with difference vectors and parametric aggregation functions?

---

> ### Author Response · Authors · 2025-11-25
>
> **“The paper doesn't discuss the computational cost of k-NN retrieval. What is the time complexity for training and inference? How does this scale with dataset size and k?”**
>
> Great point! To study the computational cost of retrieval, we conducted an empirical study of the training and inference speed with DARP (using k-NN retrieval) vs. standard BC:
> | Environment | $k$ | Train (s/epoch) | Test (s/step) |
> | :--- | :--- | :--- | :--- |
> | **Hopper (1k datapoints)** | BC | 0.102 | 0.00127 |
> | | 100 | 0.126 | 0.00413 |
> | | 200 | 0.128 | 0.00418 |
> | | 300 | 0.130 | 0.00420 |
> | | 400 | 0.131 | 0.00421 |
> | | 500 | 0.130 | 0.00419 |
> | **Stack (4.2k datapoints)** | BC | 0.431 | 0.00139 |
> | | 100 | 0.556 | 0.00437 |
> | | 250 | 0.539 | 0.00442 |
> | | 500 | 0.559 | 0.00437 |
> | | 750 | 0.576 | 0.00433 |
> | | 1000 | 0.591 | 0.00478 |
> | | 1500 | 0.682 | 0.00452 |
> | | 2000 | 0.758 | 0.00451 |
>
> 1. Training vs. Inference overhead
> The computational overhead of DARP is manageable across both training and inference. For the Stack task, using $k = 500$ only increases training time by about 30%.  On our hardware, inference time with 500 neighbors is about 0.00437 seconds per step, meaning we could run over 230 Hz, which is likely to be sufficient for nearly all real-time robotic applications, demonstrating that cost at inference time should not be prohibitive
>
> 2. Scaling with dataset size and $k$
> Empirically, we find that computation cost scales sub-linearly as $k$ increases. For instance, on the Stack task, increasing $k$ from 500 to 2000 increases training time by about 35% and inference time by only 3%. As for dataset size scaling, going from Hopper (1000 datapoints) to Stack (4200 datapoints) and keeping $k$ fixed at 500, inference time increases by only 4%. Training time isn’t directly comparable due to changes in batch size, but BC time per step increases by 323% while DARP training time increases by 330% while keeping $k$ fixed at 500, increasing nearly the same as standard BC. As for significantly larger datasets (those much larger than the ones in this paper), Approximate Nearest Neighbor such as FAISS [1, 2] can be leveraged for efficient retrieval.
>
> **“lacks comparison with other smoothness-inducing approaches mentioned in related work“**
>
> The authors of CCIL [3] benchmarked against the same MuJoCo tasks that we performed, so we can compare our relative improves compared to standard behavior cloning:
>
>
> | Environment | Method | Improvement (over BC) |
> | :--- | :--- | :--- |
> | **Hopper** | CCIL | 32.6% |
> | | DARP | 53.2% |
> | **Walker** | CCIL | 84.1% |
> | | DARP | 84.1% |
> | **Ant** | CCIL | 12.6% |
> | | DARP | 84.5% |
> | **HalfCheetah** | CCIL | 5.4% |
> | | DARP | 418.7% |
>
> We see that DARP offers higher gains on all but one task, Walker, on which the two are equal in their improvement.
>
> **“How sensitive is DARP to the choice of distance metric?”**
>
> To answer this question, we have experimented with moving from Euclidean distance to the cosine similarity metric, particularly when using the high-dimensional R3M features. We found that it performs similarly, getting a success rate of 70% versus our reported 75% with Euclidean distance, both handily beating BC’s 47%. However, this is still an open and active area of research for us, and experimenting with learned metrics would be interesting future work.
>
> **“Does the Laplacian smoothing analysis extend to the full DARP algorithm with difference vectors and parametric aggregation functions?”**
> While the analysis does not hold directly for full difference vectors and parametric aggregation functions, we justify this choice through careful experimental analysis:
> We carefully perform an experimental analysis showing that difference vectors are crucial in ensuring that new points are in-distribution, as shown in Fig 5 and Section 3.4.
> We perform several of our experiments using averaging based aggregation rather than learned aggregation, where the analysis will hold directly.
> We agree that a theoretical study of the entire algorithm is an exciting direction of future work, and we are keen to pursue this in future work.
>
> [1] Douze, Matthijs, et al. "The Faiss library.(2024)." arXiv preprint arXiv:2401.08281 (2024).
>
> [2] Johnson, Jeff, Matthijs Douze, and Hervé Jégou. "Billion-scale similarity search with GPUs." IEEE Transactions on Big Data 7.3 (2019): 535-547.
>
> [3] Liyiming Ke, Yunchu Zhang, Abhay Deshpande, Siddhartha S. Srinivasa, and Abhishek Gupta. CCIL: continuity-based data augmentation for corrective imitation learning. In The Twelfth International Conference on Learning Representations, ICLR 2024, Vienna, Austria, May 7-11, 2024. OpenReview.net, 2024b. URL https://openreview.net/forum?id=LQ6LQ8f4y8.

---

### Author Response · Authors · 2025-12-03
**Summary for Area Chair**

Hello Area Chair,

To aid you in the review process, we summarize below the concerns/questions of all reviewers and how we addressed them during the rebuttal period. We believe we have addressed all major concerns posed by our reviewers through new experimental results, improved mathematical clarity throughout the paper, and thorough responses otherwise. We thank the reviewers for their time and helpful suggestions – we believe that our work has been considerably improved during this discussion period.

**Computational Cost**

Reviewer T2hb and kF51 expressed concern regarding the computational cost of our method, which is an absolutely reasonable concern. In response, we provided a thorough analysis of computation cost at training and inference time across various values of $k$ and sizes of dataset (see Table 12). We demonstrated that our method can run inference at approximately 0.00437 seconds per step at $k = 500$ on a dataset with 4,200 datapoints, corresponding to a control frequency higher than 230 Hz – this is more than fast enough to apply to nearly all real-time robotic applications. Additionally, we find that computation cost scales sub-linearly as $k$ increases. See section A.4 for more details.

**Additional Baselines**

Reviewers T2hb and vWKv both requested additional baselining to other methods. Reviewer T2hb requested a comparison to another smoothness-inducing approach, so we compared our numbers with the numbers published on the same tasks by the CCIL (T2hb response, citation [3]) authors to show that our method outperforms CCIL on MuJoCo baselines (Table 6 and section A.2.4). On one of the environments, DARP improves upon BC by ~419%, while CCIL only improves upon BC by ~5%. Reviewer vWKv requested additional benchmarking against diffusion policies for manipulation tasks. We demonstrated that our method does outperform state-of-the-art diffusion policies and is additionally about 10 times faster in training and 36 times faster in inference (see Table 9). We also shared additional results demonstrating that DARP architecture is fully compatible with diffusion techniques, and that the combination of the two leads to even better results, which reveals an exciting new research direction for our method. This is demonstrated in table 8 and section A.2.7, where we show that DARP with a diffusion backbone outperforms DARP with an MLP backbone by 14% and outperforms standard BC by 42%.

**Performance on Human Data**

Reviewer iFBA desired results demonstrating that our method outperforms baselines when trained on human expert demonstrations, as opposed to the RL-policy demonstrations we train on in our paper. We demonstrate that this is indeed the case in Table 10 and section A.2.8.

**Distance Metrics**

Reviewers T2hb and kF51 questioned our use of Euclidean distance in high-dimensional embeddings (like the R3M feature-based experiments we include in our submission). To address this, we empirically demonstrated that the use of cosine similarity rather than Euclidean distance yielded very similar results in high-dimensional embeddings, beating BC by 23% and 28% respectively. See Table 7 and section A.2.6 for more details.

**Choice of Retrieval Hyperparameters**

vWKv pointed out that some choices of retrieval hyperparameters cause our method to perform worse than baselines. To address this, we demonstrated that using the metric of validation loss to select retrieval hyperparameters is a cost-effective way to isolate sets of parameters that indeed cause our method to outperform baselines. See Table 11 and section A.2.9 for more details.

**Mathematical Clarity**

Reviewers kF51 and iFBA rightfully asked about some mathematical symbols in our work that were presented without definitions ($s^\star$ and $a^\star$ on page 2, $L_a$, $L_s$, and $\pi(s_t)$ on page 4, and $r$ on page 16). We have edited our paper to include definitions for these symbols, and thank the reviewers for their pointers.

**Divergence Analysis Confusion**

Reviewer vWKv had concerns regarding our divergence analysis (Figure 5), but we are concerned that they may have misunderstood this section. We included this analysis to empirically analyze post-hoc the exact point of divergence between standard BC and our method during a rollout, but want to emphasize that the detection of this divergence is not at all a requirement for our method to work. We hope our response quells any concerns regarding this analysis.

**Miscellanea**

In addition to the above, we addressed reviewer T2hb’s theoretical concerns regarding Laplacian smoothing, clarified the generalization performance of DARP for reviewer vWKv, and we compared and contrasted our method to a paper recommended by reviewer iFBA entitled “Online Adaptation for Enhancing Imitation Learning Policies,” and clarified the motivation behind the use of Set Transformers for reviewer iFBA.

---

### Meta-Review · Area_Chair_siA8 · 2026-01-05

**Summary:**

This paper proposes DARP, which integrates non-parametric retrieval with parametric behavior cloning for offline imitation learning. The action predictions are based on k-nearest neighbors from the expert demonstrations, particularly the relative distance between neighbor states and query states.


Strengths:
- The idea of reparameterizing the policy based on local neighborhood structure and leveraging retrieval is novel

- Both theoretical and empirical analysis are good

Weaknesses:
- The computation cost including both time and memory is relatively high

- The paper can be improved by providing a comprehensive comparison with diffusion-based BC on manipulation tasks

**Reviewer Concerns:**

The rebuttal did clarify most questions raised by reviewers, including computational cost comparison (reviewers T2hb, kF51), baseline comparison (reviewer T2hb), comparison with diffusion-based BC on manipulation tasks (reviewer vWKv), questions about distance metrics (reviewers T2hb, vWKv, kF51), and some clarification issues.

However, due to the nature of the algorithm design with non-parametric methods, the computation cost of the proposed algorithm is relatively high. Besides, while the initial comparison with diffusion-based BC seems promising on manipulation tasks, this paper can be further strengthened by providing a more comprehensive evaluation on this.

**Reviewer Scores:**

Reviewers T2hb, kF51, iFBA may maintain their scores as they were positive initially while being aware of the potentially high computational cost.

Reviewer vWKv may increase the score to 6, because the overall evaluation of the paper from the reviewer was good and the additional experimental results provided in the rebuttal were helpful to address the major concerns.

---

### Decision · Program_Chairs · 2026-01-26

Accept (Poster)